# Improving Equivariant Networks with Probabilistic Symmetry Breaking

**Hannah Lawrence**[1]*, **Vasco Portilheiro**[2]*, **Yan Zhang**[3,4], **Sékou-Oumar Kaba**[4,5]
[1]Department of Electrical Engineering and Computer Science, MIT,
[2]Gatsby Computational Neuroscience Unit, UCL, [3]Samsung – SAIT AI Lab, Montreal,
[4]Mila – Quebec Artficial Intelligence Institute, [5]McGill University

## Abstract

Equivariance encodes known symmetries into neural networks, often enhancing generalization. However, equivariant networks cannot *break* symmetries: the output of an equivariant network must, by definition, have at least the same self-symmetries as the input. This poses an important problem, both (1) for prediction tasks on domains where self-symmetries are common, and (2) for generative models, which must break symmetries in order to reconstruct from highly symmetric latent spaces. This fundamental limitation can be addressed by considering *equivariant conditional distributions*, instead of equivariant functions. We present novel theoretical results that establish necessary and sufficient conditions for representing such distributions. Concretely, this representation provides a practical framework for breaking symmetries in any equivariant network via randomized canonicalization. Our method, SymPE (Symmetry-breaking Positional Encodings), admits a simple interpretation in terms of positional encodings. This approach expands the representational power of equivariant networks while retaining the inductive bias of symmetry, which we justify through generalization bounds. Experimental results demonstrate that SymPE significantly improves performance of group-equivariant and graph neural networks across diffusion models for graphs, graph autoencoders, and lattice spin system modeling.

## 1 Introduction

Learning tasks with known symmetries, such as rotations and permutations, abound in applications (Veeling et al., 2018; Celledoni et al., 2021; Bogatskiy et al., 2022; Veličković, 2023). Equivariant learning, which builds these symmetries directly into neural networks, has been shown to provide a powerful inductive bias for deep learning (Bronstein et al., 2021). However, even in domains that seemingly have clear symmetries, there are functions that equivariant networks simply cannot represent. For example, consider the problem of predicting one molecular three-dimensional graph from another, such as predicting a dichlorobenzene molecule from a benzene molecule (pictured at the top of Fig. 1). Such tasks are relevant in generative modeling of atomic systems (Satorras et al., 2021; Xie et al., 2021) and molecular editing (Liu et al., 2024). Since we are working in 3D space, rotation equivariance is a natural choice—intuitively, rotating the benzene molecule should only affect the rotation of the predicted dichlorobenzene, not the structure of the molecule itself.

While this approach seems reasonable, a strange problem arises. When the input to an equivariant model is self-symmetric, it *must* remain self-symmetric in the output (as pointed out by e.g. Smidt et al. (2021)). Since benzene has sixfold rotational symmetry, an equivariant model is *unable* to output dichlorobenzene, which is not rotationally symmetric.

In fact, self-symmetry arises in a variety of applications, often with more complex groups—e.g. non-trivial graph automorphisms, Hamiltonians of physical systems with symmetries, or rotationally symmetric point-clouds (Fig. 1). Generative models and autoencoders, which reconstruct from a latent space, are particularly noteworthy. By virtue of being embedded in a simple, low-dimensional space, the latent representation often has greater self-symmetry than the input itself, i.e. certain

---

*Author order decided by coin flip. Correspondence to: `hanlaw@mit.edu`, `vascopo@gmail.com`
Code available at: `https://github.com/hannahlawrence/symm-breaking`

transformations of the input will not affect its latent representation. An equivariant decoder must then map the *more* symmetric latent space to the *less* symmetric data space, which is just as impossible as predicting dichlorobenzene from benzene. To avoid this problem, we could simply discard symmetry structure entirely, but this loses the generalization benefits of equivariance on asymmetric inputs. The question we ask is therefore: *How can we retain the inductive bias of symmetry, while resolving the difficulty posed by self-symmetric inputs?*

We will focus on equivariant *distributions*, i.e. functions from the input space to distributions over the output space, rather than the more standard setting of equivariant functions from input to output space. This provides greater flexibility in the types of systems we can model, because as we will see later, a *distribution* can be equivariant without *individual samples* from that distribution transforming equivariantly. Instead of mapping an input to a single output through an equivariant function directly, we map the input to an equivariant distribution, and *sample* from it. This distinction will allow us to resolve the symmetry breaking problem elegantly.

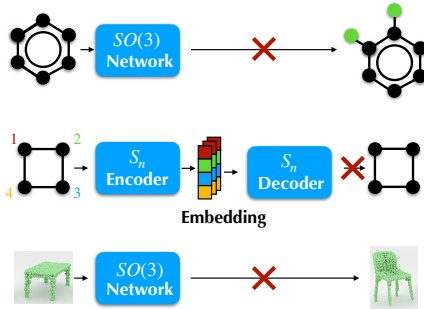

Figure 1: Example applications requiring symmetry breaking. *Top*: A rotation-equivariant network for molecules cannot transform benzene into dichlorobenzene due to benzene's sixfold symmetry. *Middle*: A permutation-equivariant graph decoder cannot break latent space symmetries (see Appendix H). *Bottom*: A rotation-equivariant network for point clouds cannot transform a table into a chair, as the table's legs are rotationally indistinguishable, unlike the chair's.

**Contributions** Extending the results of Bloem-Reddy & Teh (2020) on probabilistic symmetries, we derive a method which can provably represent *any* equivariant conditional distribution (Section 3). For this, we rely on an external source of randomness, coming from a canonicalization function which has been appropriately randomized. Our theory suggests a simple modification to existing equivariant models, akin to concatenating a positional encoding, which allows them to break symmetries (Section 4). In Section 5, we show more generally that equivariant noise injection can break symmetries, and provide theoretical justification for its generalization benefits in Section 6. Using our framework, we also show in Section 7 and Corollary A.3 that several recently proposed approaches to symmetry breaking (Kaba & Ravanbakhsh, 2023; Xie & Smidt, 2024) can be modified to represent any equivariant conditional distribution. Finally, we validate our approach experimentally in tasks on graphs, atomic systems, and spin Hamiltonians (Section 8).

## 2 BACKGROUND

**Preliminaries** Let $\mathcal{X}$ and $\mathcal{Y}$ be measurable input and output spaces, respectively. We assume $G$ is a group which acts on both $\mathcal{X}$ and $\mathcal{Y}$, with the action of $g \in G$ on $x \in \mathcal{X}$ denoted by $gx \in \mathcal{X}$.[1] Equivariance describes functions $f$ satisfying $f(gx) = gf(x)$, capturing the requirement that the output should transform predictably under transformations of the input. Given a subgroup $H \subseteq G$, we denote a *left coset* of $H$ in $G$ as $gH \equiv \{gh : h \in H\}$ for $g \in G$. We denote *right cosets* similarly as $Hg$. The *orbit* of $x \in \mathcal{X}$ with respect to $G$ is $[x] \equiv \{gx : g \in G\}$. The *stabilizer* of $x$ is denoted by $G_x \equiv \{g \in G : gx = x\}$ and is the subset of $G$ that leaves $x$ unchanged. Any $x$ with non-trivial $G_x$ is termed *self-symmetric*. We consider probability distributions over $\mathcal{X}$, $\mathcal{Y}$, and $G$, with $\mathcal{P}(D)$ denoting the space of all probability distributions over $D$. Also, denote by $X$ and $Y$ the input and output random variables, respectively. For random variables $X_1$ and $X_2$, we write $X_1 =_{a.s.} X_2$ if the equality holds with probability 1. Denote the distribution of $X$ by $\mathbb{P}(X)$, and the conditional distribution of $Y$ given $X$ by $\mathbb{P}(Y|X)$. The action of $G$ on $\mathcal{Y}$ naturally gives rise to an action on *distributions*, defined by $g \cdot \mathbb{P}(Y) \equiv \mathbb{P}(gY)$ as shown in Figure 2. Essentially, the action of $g$ on $\mathbb{P}(Y)$ is the distribution of the random variable $Y$ after transformation by $g$.

Following Bloem-Reddy & Teh (2020), the definition of equivariance can be extended to conditional distributions $\mathbb{P}(Y|X)$ simply by viewing $\mathbb{P}(Y|X)$ as a function from $X$ to $\mathcal{P}(Y)$, and applying the standard definition of equivariance to the action of $G$ on $\mathcal{P}(Y)$. In essence, transforming the value of

---

[1]$G$ should be locally compact, second countable Hausdorff, with *proper* actions on $\mathcal{X}$ and $\mathcal{Y}$ (Chiu & Bloem-Reddy, 2023).

the input random variable $x \in \mathcal{X}$ by $g$ just shifts the distribution of $Y$ by $g$:

$$\mathbb{P}\left(Y|X=gx\right) = g \cdot \mathbb{P}\left(Y|X=x\right) = \mathbb{P}\left(gY|X=x\right). \tag{1}$$

**Curie's Principle**    As discussed previously, the output of equivariant functions must be at least as self-symmetric as the input. This is known as Curie's Principle in physics (Curie, 1894). It can be stated formally in terms of stabilizers as $G_x \subseteq G_{f(x)}$ for any $x \in \mathcal{X}$ if $f$ is an equivariant function. The proof is quite simple: if $g \in G_x$, then $x = gx$ implies $f(x) = f(gx) = gf(x)$.

If we consider a *probabilistic* system, Curie's Principle holds at the level of *distributions* instead. That is, if $\mathbb{P}\left(Y|X\right)$ is equivariant, then for a given $x$, the *distribution* $\mathbb{P}\left(Y|X=x\right)$ inherits the same self-symmetries as $x$: $G_x \subseteq G_{\mathbb{P}(Y|X=x)}$. A key insight is that this does not need to hold for individual samples from $\mathbb{P}\left(Y|X=x\right)$, as shown in Figure 3. This is known by physicists as *spontaneous symmetry breaking* (Beekman et al., 2019). In this work, we posit that symmetry breaking is best understood as coming from the probabilistic nature of the world.

While equivariant distributions provide an elegant way of addressing symmetry breaking, equivariant modeling is usually done with deterministic networks. We will now see how to bridge the two.

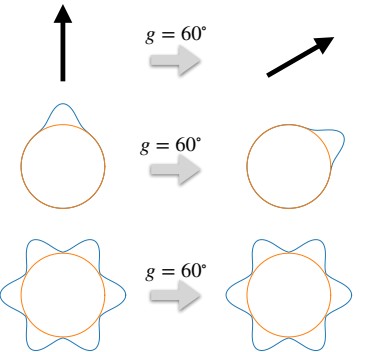

Figure 2: Example of how a group acts on distributions. *Top:* An element $g \in SO(2)$, the group of two-dimensional rotations, can act on a unit vector in $\mathbb{R}^2$ by standard rotation. *Middle:* This induces an action of $g$ on a *distribution* (blue) over unit vectors (orange). Under this action, $g$ rotates the entire distribution. *Bottom:* When $g = 60°$ acts on a distribution which is already $60°$ self-symmetric, the distribution remains unchanged—even though a vector sampled from the distribution has no $60°$ self-symmetry.

**From equivariant functions to distributions**    Bloem-Reddy & Teh (2020) link together the two concepts by expressing equivariant distributions in terms of equivariant functions. They show that under certain conditions,[2] $\mathbb{P}\left(Y|X=x\right)$ is equivariant if and only if there exists a function $f : \mathcal{X} \times (0,1) \to \mathcal{Y}$ equivariant in $x$ (i.e. $f(gx, \epsilon) = gf(x, \epsilon)$) such that:

$$Y \stackrel{a.s.}{=} f(X, \epsilon) \tag{2}$$

with $\epsilon \sim \mathrm{Unif}(0,1)$ a random variable independent of $X$. However, one necessary condition is that $G$ acts on $\mathcal{X}$ *freely*: each $x \in \mathcal{X}$ must not have any self-symmetries (i.e. the stabilizer $G_x$ is trivial). Therefore, this result does not apply to the kinds of inputs for which we would like to break symmetries. In the following, we show how to address this limitation.

## 3    REPRESENTATION OF EQUIVARIANT DISTRIBUTIONS

Our aim is now to write equivariant distributions in terms of equivariant functions, while handling possibly self-symmetric inputs. The main idea is to introduce a mapping called the *inversion kernel* (Kallenberg, 2011). Intuitively, the inversion kernel maps an input $x$ to a uniform distribution over a subset of group elements describing the input's self-symmetry. Sampling from the inversion kernel will then let us break the self-symmetry of the input, and ultimately represent equivariant distributions in terms of equivariant functions.

To formally define the inversion kernel, we first use the concept of canonicalization (Kaba et al., 2023). Canonicalization allows for transforming any input to a canonical "pose," or *orbit representative*.

**Definition 3.1** (Canonicalization function). A function $\tau : \mathcal{X} \to G$ is a *canonicalization function* if $\gamma(x) \equiv \tau(x)^{-1}x$ is invariant for all $x \in \mathcal{X}$. The function $\gamma : \mathcal{X} \to \mathcal{X}$ is the *orbit representative map* associated with the canonicalization function, and $\gamma(x)$ is the *orbit representative*.

Invariance of the orbit representative map ensures that each transformed version of $x$ gets assigned to a single representative of the orbit of $x$. The canonicalization function can alternatively be seen as

---

[2]$G$ is compact, $\mathbb{P}\left(X\right) = g \cdot \mathbb{P}\left(X\right)$, and there is a measurable canonicalization map (as defined in Section 3).

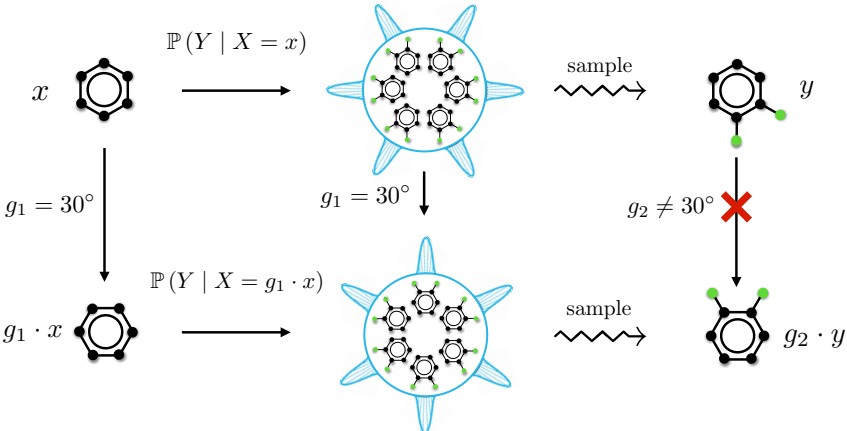

Figure 3: When the input $x$ is rotated by $30°$, as shown from the top to the bottom row, the equivariant conditional distribution $\mathbb{P}(Y|X=x)$ (middle column) also rotates by $30°$. The *distribution* thus has the same self-symmetry as $x$, which is sixfold rotational symmetry. However, individual samples from $\mathbb{P}(Y|X=x)$ are free to break this self-symmetry, as shown in the rightmost column.

providing a group element which maps the orbit representative to a given input, i.e. $x = \tau(x)\gamma(x)$. In the absence of self-symmetries, the canonicalization function is uniquely defined (and equivariant), given $\gamma$. But consider the case where $x$ has a self-symmetry. For $g \in G_x$, we have $gx = x$ and therefore $x = g^{-1}x$, and so for any canonicalization $\tau(x)$ taking $x$ to $\gamma(x)$, it holds that

$$(g\tau(x))^{-1}x = \tau(x)^{-1}g^{-1}x = \tau(x)^{-1}x = \gamma(x).$$

The entire coset $G_x\tau(x)$ comprises the possible ways to get to $x$ from $\gamma(x)$. The inversion kernel is a map from $x$ to a uniform distribution over these transformations that canonicalize $x$ in the same way.

**Definition 3.2** (Inversion kernel). Let $\tau : \mathcal{X} \to G$ be a canonicalization function. The corresponding *inversion kernel*, mapping $x$ to $\mathcal{P}(G)$, is the conditional distribution $\mathbb{P}(\tilde{g}|X=x) = \text{Unif}(G_x\tau(x))$.

**Example 3.3.** For example, consider $G = SO(2)$, and let $x$ be the benzene ring at upper left of Fig. 3. If we set $\tau(x) = 30°$ (such that the orbit representative $\gamma(x)$ is at the lower left), then $G_x = \{0°, 60°, \ldots, 300°\}$, and the inversion kernel at $x$ is uniform over $G_x\tau(x) = \{30°, 90°, \ldots, 330°\}$.

While each choice of canonicalization map $\tau$ gives rise to an inversion kernel, we often refer to "the" inversion kernel for simplicity, assuming tacitly some $\tau$ has been chosen (and is universally measurable (Kallenberg, 2017)). Using this, we can generalize Bloem-Reddy & Teh (2020) to represent *any* equivariant distribution.

**Theorem 3.4.** $\mathbb{P}(Y|X)$ *is equivariant if and only if*

$$Y \stackrel{a.s.}{=} f(X, \tilde{g}, \epsilon) \tag{3}$$

*for a function $f : \mathcal{X} \times G \times (0,1) \to \mathcal{Y}$ jointly equivariant in its first two inputs (i.e. $f(hx, hg, \epsilon) = hf(x, g, \epsilon)$), noise $\epsilon \sim \text{Unif}(0, 1)$, and $\tilde{g}|X$ distributed according to some inversion kernel.*

The proof follows in Appendix A, where we also prove a related representation in terms of *relaxed equivariant* functions (Kaba & Ravanbakhsh, 2023). Intuitively, our result decomposes the randomness in $Y|X$ into that derived from symmetry breaking, $\tilde{g}$, and independent noise $\epsilon$. Here, $\tilde{g}$ selects among various transformations $g \in G$ which canonicalize $X$, and since any group element has a trivial stabilizer, this allows $Y$ to break self-symmetry; in essence, we bootstrap the symmetry breaking of existing canonicalization techniques to that of $\mathbb{P}(Y|X)$. Note that in this work, we are primarily concerned with the randomness deriving from symmetry breaking and not learning generic equivariant distributions, and so we will focus primarily on $\tilde{g}$, omitting $\epsilon$ from experiments.

## 4  METHOD: SYMMETRY-BREAKING POSITIONAL ENCODING (SYMPE)

Following Theorem 3.4 and as shown in Fig. 4, we propose to represent equivariant conditionals using an equivariant neural network $f$, and pass in $(x, \tilde{g})$ as input (and $\epsilon$, if so desired). Two implementation questions remain: (1) how to sample $\tilde{g}$, and (2) how to pass $\tilde{g}$ as input to an equivariant network $f$. We first introduce a general approach to sample from inversion kernels using canonicalization.

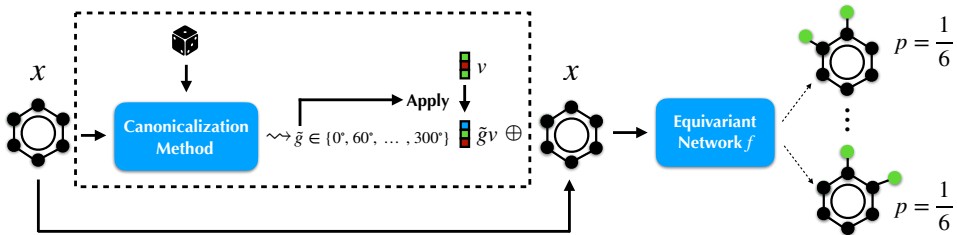

Figure 4: Illustration of our symmetry-breaking method. Here, a die indicates randomness, which is used in the canonicalization method (shown in the dotted box) to sample $\tilde{g}$. (Optionally, a random variable $\epsilon$ can also be input to the equivariant network $f$, to capture randomness unrelated to symmetry-breaking; we do not include this variable in our experiments.) Ultimately, the input $x$ and the sampled group element $\tilde{g}$ are input to an equivariant network $f$ as $f(x, \tilde{g})$.

Then, we show how to encode the group element $\tilde{g}$ as input to the neural network. We finally provide an interpretation of our method as a positional encoding (Vaswani et al., 2017; Bello et al., 2019; You et al., 2019; Srinivasan & Ribeiro, 2020; Lim et al., 2023a). As we will see, an advantage of our method is that all the components can be learned end-to-end, and do not require much hand-engineering beyond specifying which equivariant neural network to use.

**Inversion kernels with canonicalization**    Our method requires sampling $\tilde{g} \sim \mathrm{Unif}(G_x \tau(x))$ for some choice of canonicalization function $\tau$. This can be implemented in a simple way by taking $\tau$ as a canonicalization network. The form of the distribution $\mathrm{Unif}(G_x \tau(x))$ seems to suggest that we need to detect the stabilizer of $x$ along with using a canonicalization function, but this is in fact not necessary. Following Kaba et al. (2023), an inversion kernel can be implemented by optimization of an *energy function* $E : \mathcal{X} \to \mathbb{R}$ such that that $G_x \tau(x) = \arg\min_{g \in G} E(g^{-1} x)$, where the $\arg\min$ is a set. Sampling is done simply by selecting a random element of the $\arg\min$ set. As we show in Appendix C, it is often possible to parameterize $E$ such that the optimization is fast and poses no significant overhead. For example, sorting breaks permutation symmetries, and can be viewed as an optimization of an energy function (Blondel et al., 2020). Implementations of these canonicalization functions exist for images, sets, graphs and point clouds (Mondal et al., 2023; Kim et al., 2023).

As explained in Appendix C and justified in Section 5, our method also works even when $\tilde{g}$ is sampled from an equivariant distribution with support larger than that of the inversion kernel. This arises when working with graph inputs, for example, since canonicalizing combinatorial graphs is likely not possible in polynomial time (Babai & Luks, 1983). On the other hand, when using an invariant loss, samples of $\tilde{g}$ can be replaced with a deterministic canonicalization $\tau(x)$ with no impact on the loss.

**Encoding of the group element**    Once a group element is sampled, we need to specify a way to feed it into an equivariant neural network. Our method is based on the observation that if the group $G$ acts freely on a vector $v$, then the mapping $g \mapsto gv$ is injective. The group element is therefore uniquely specified by the way it transforms $v$. We can furthermore make $v$ learnable. We show in Appendix D that vector spaces with that property can be defined conveniently for group representations of finite groups and Lie groups, which comprise most groups of interest in applications. The encoded group element is then given as input to the equivariant neural network by concatenation with $x$. Linking back to Theorem 3.4, our model is defined as $f(x, \tilde{g}) = f_0(x \oplus \tilde{g}v)$, where $f_0$ is any equivariant neural network. The equivariance of $(x, g) \mapsto x \oplus (gv)$ is clear, so $f$ is indeed a jointly equivariant function. As shown in Appendix D, any jointly equivariant function $f$ can be expressed in this way, which makes our method fully expressive.

**Interpretation as positional encodings**    We term our method *Symmetry-breaking Positional Encoding* (SymPE), since it can be naturally interpreted as a type of positional encoding, similar to the ones used in transformers (Vaswani et al., 2017) and GNNs (You et al., 2019). In general, absolute positional encodings can disambiguate between identical tokens or nodes by assigning each a unique identifier. In graphs, for example, nodes which are part of the same orbit of the automorphism group (and that are thus indistinguishable for an equivariant model) are assigned different "positions" by the positional encoding. The positional encoding therefore breaks symmetries in the input. We generalize this view to other data types and groups. Specifically, the sampled group element $\tilde{g}$ represents the position (for translation groups), pose (for rotation groups) or ordering (for permutation groups)

---

**Algorithm 1** SymPE: Symmetry-Breaking Positional Encodings

---

1: **Inputs:** input $x \in \mathcal{X}$
2: **Learnable parameters:** learned vector $v \in \mathcal{V}$, equivariant neural network parameters $\theta$, canonicalization parameters $\phi$
3: Sample $\tilde{g} \sim h_\phi(x)$  ▷ Sample group element for canonicalization
4: $\tilde{v} \leftarrow \tilde{g}v$  ▷ Apply group element to learned vector
5: Return $f_\theta(x \oplus \tilde{v})$  ▷ Forward pass with positional encoding

---

of the input relative to a canonical one. We apply this group element to a learned vector $v$ that is concatenated to each component of the input (token, pixel, node, etc.). By virtue of the group acting freely on $v$, it fully specifies the "position" of the input. The learned vector therefore plays the same role as the sinusoidal encoding in sequence models, with the group element $\tilde{g}$ shifting these encodings to specify which token is the first one.

**Equivariance and symmetry-breaking of positional encodings**  The standard, absolute positional encodings used in Transformers (Vaswani et al., 2017) are not equivariant. In other words, the position nominally assigned to tokens in the sequence by the positional encoding does not shift if the sequence shifts, which precludes translation equivariance. Similarly, the pixelwise positional encodings used in Vision Transformers do not allow translation equivariance (in contrast to CNNs). Relative positional encodings are equivariant to shifts (Shaw et al., 2018), but cannot break symmetries because they rely only on invariant relative distances between tokens. The Laplacian positional encodings used in GNNs (Belkin & Niyogi, 2003; Dwivedi et al., 2023), are also not equivariant to permutations, but can be made so at the cost of losing the ability to break automorphism symmetries (Lim et al., 2023c; Morris et al., 2024). The same is true for positional encodings based on random walks (Dwivedi et al., 2022; Ma et al., 2023), which preserve automorphisms. By contrast, in our method, the absolute position $\tilde{g}$ is sampled from the inversion kernel, which is an equivariant distribution. As a result, the model preserves the inductive bias of equivariance, while retaining the ability to differentiate between different symmetric configurations and therefore break symmetries.

## 5 SYMMETRY BREAKING WITH NOISE INJECTION

Adding noise to inputs is a commonly used heuristic for symmetry breaking (Satorras et al., 2021; Sato et al., 2021; Abboud et al., 2021; Eliasof et al., 2023; Zhao et al., 2024). This approach involves using a functional model similar to that of Theorem 3.4, but with an arbitrary equivariant noise variable $Z$ replacing $\tilde{g}$. It is natural to ask whether this simple heuristic can also represent any equivariant distribution. We show that under some conditions, noise injection can indeed be used to break symmetries, while still representing any equivariant conditional distribution.

**Proposition 5.1** (Noise injection). *Let $X, Y, Z$ be random variables in $\mathcal{X}, \mathcal{Y}, \mathcal{Z}$ respectively, each space acted on by $G$. The following are equivalent: (1) $G$ acts on $\mathcal{Z}$ freely (up to a set of probability zero) and $\mathbb{P}(Z|X)$ is equivariant; (2) $\mathbb{P}(Y|X)$ is equivariant iff there exists $f : \mathcal{X} \times \mathcal{Z} \times (0,1) \to \mathcal{Y}$ jointly equivariant in $X$ and $Z$ such that $Y \overset{a.s.}{=} f(X, Z, \epsilon)$ for noise $\epsilon \sim \text{Unif}(0,1)$.*

The proof follows in Appendix A.4. This result implies that one may sample $\tilde{g}$ from a general equivariant distribution on $G$ instead of the inversion kernel specifically, and still obtain an equivariant conditional distribution.[3] This applies, for example, to the recently proposed *symmetry breaking sets* (SBS) of Xie & Smidt (2024) (see Section 7). Moreover, for many groups of interest, such as $S_n$ and $O(n)$, simply sampling $Z$ from an isotropic Gaussian satisfies the requirements of Proposition 5.1. However, this method potentially introduces more noise than is necessary into the learning process. More precisely, as we show in Appendix E, the inversion kernel is an equivariant distribution of minimal entropy, which we conjecture facilitates learning (since the model need not learn to map more random inputs to the same output than necessary). Our ablation studies, comparing the inversion kernel to a generic distribution, align with this intuition (Section 8).

---

[3]In fact, the optimization of energy mentioned above is akin to sampling from a density $p(g|x) \propto \exp(-E(g^{-1}x)/T)$. When $T > 0$ we sample from a general equivariant distribution on $G$. As we lower $T$, the entropy diminishes and we recover an exact inversion kernel $\arg\min_{g \in G} E(g^{-1}x)$ when $T \to 0$.

## 6 GENERALIZATION BENEFITS OF SYMMETRY BREAKING

Equivariance is understood to impart generalization benefits by meaningfully restricting the hypothesis class. In this section, we explore similar intuitions for symmetry-breaking.

Elesedy & Zaidi (2021) formalized this intuition for ordinary equivariance. They showed that if $\mathbb{P}(X)$ is $G$-invariant, then the space $L^2(\mathcal{X}, \mathcal{Y}, \mathbb{P}(X))$ of square-integrable functions $f$, with inner product $\langle f_1, f_2 \rangle_{\mathbb{P}(X)} = \int \langle f_1(x), f_2(x) \rangle_\mathcal{Y} \mathbb{P}(dx)$, decomposes into the orthogonal sum of equivariant and "anti-equivariant" parts, $\bar{f}$ and $f^\perp = f - \bar{f}$. Assuming $Y = f^*(X) + \epsilon$ for an equivariant function $f^*$ and mean-zero noise $\epsilon$, it follows that for $L^2$ risk $R(f) = \mathbb{E}[\|f(X) - Y\|^2]$, there is a non-negative *generalization gap* $\Delta(f, \bar{f}) = R(f) - R(\bar{f}) = \|f^\perp\|^2_{\mathbb{P}(X)}$ when using a non-equivariant model $f$. This provides some theoretical justification for the use of equivariant models: it implies that for any non-equivariant model $f$, there exists an equivariant model of strictly lower risk. In Theorem 6.1 (proved in Appendix A.5), we obtain a similar result for our probabilistic setting, simply assuming $\mathbb{P}(Y|X)$ is equivariant and that one is injecting equivariant noise.

**Theorem 6.1.** *Suppose $\mathbb{P}(X)$ is $G$-invariant, and $\mathbb{P}(Y|X)$ is equivariant with $\mathbb{E}[Y^2] < \infty$. Consider a stochastic model $\hat{Y} = f(X, Z)$ with $\mathbb{E}[\hat{Y}^2] < \infty$, where $Z \in \mathcal{Z}$ is such that $\mathbb{P}(Z|X)$ is equivariant, and where $G$ acts on $\mathcal{Z}$ freely (up to a set of zero probability). Then $f$ decomposes into jointly equivariant $\bar{f}$ and its orthogonal complement $f^\perp = f - \bar{f}$, and there is a generalization gap*

$$\Delta(f, \bar{f}) = R(f) - R(\bar{f}) = \|f^\perp\|^2_{\mathbb{P}(X,Z)}, \tag{4}$$

*where $R(f) = \mathbb{E}[\|f(X, Z) - Y\|^2]$.*

**Remark 6.2.** The condition that $\mathbb{P}(Z|X)$ is equivariant is satisfied when $\mathbb{P}(Z)$ is simply a $G$-invariant distribution independent of $X$, which is often the case. The result above says that even in this pure noise injection setting, the expected loss of a model $f(X, Z)$ can be decreased by projecting $f$ to an equivariant function, thereby producing a conditionally equivariant distribution. The result also applies to SymPE, by letting $Z = (\tilde{g}, \epsilon)$ as in Theorem 3.4, where $g$ acts as $gZ = (g\tilde{g}, \epsilon)$.

We also emphasize that the risk used in the theorem measures error in each individual prediction of $Y$ from $X$. This may not be the metric of interest, if, for example, one only cares about predictions up to a group transformation, or wants to learn the entire distribution for a generative task. In Appendix B, we prove a similar result when instead computing the loss between *distributions*, i.e. between a ground truth equivariant conditional density and an arbitrary model density.

## 7 RELATED WORK

The problem of self-symmetric inputs yielding self-symmetric outputs in equivariant learning has been observed in several domains. In the context of graph representation learning, Srinivasan & Ribeiro (2020) showed that isomorphic structures in graphs must be assigned the same representations by equivariant functions. It has been noted that this is problematic in several applications, including generative modeling on graphs (Liu et al., 2019; Satorras et al., 2021; Yan et al., 2023; Zhao et al., 2024) and link prediction (Lim et al., 2023b). Similar issues arise for discriminative and generation tasks on sets. Zhang et al. (2022) noted that equivariant functions are limited in both processing multisets (sets with self-symmetries) and decoding from an invariant latent space to a set, introducing a notion of *multiset equivariance*, of which *relaxed equivariance* (Kaba et al., 2023; Kaba & Ravanbakhsh, 2023) is a generalization, as a solution. Vignac & Frossard (2022) also introduced a generalization of equivariance to address the problem of set generation, which our results subsume.

Other studies have focused on symmetry breaking in the context of machine learning for modeling physical systems. Smidt et al. (2021) first formulated the preservation of symmetry in equivariant neural networks as an analogue to Curie's Principle, and proposed using gradients of equivariant neural networks to identify cases for which symmetry breaking is necessary. Kaba & Ravanbakhsh (2022) identified that for prediction tasks on crystal structures, symmetry breaking can be necessary, and proposed a method based on *non-equivariant* positional encodings. For inputs defined on discrete grids, Wang et al. (2024) proposed a flexible method for implementing and interpreting symmetry breaking based on *relaxed group convolutions*, at the cost of only approximating equivariance.

On the theoretical side, our work builds on that of Chiu & Bloem-Reddy (2023), who appear to have been first in applying the inversion kernel in machine learning. Sampling from a general equivariant

conditional distribution on $G$ was also explored by Dym et al. (2024), in the context of constructing continuous and efficient frames (Puny et al., 2022). Symmetry breaking more specifically was studied by Kaba & Ravanbakhsh (2023), who defined the notion of relaxed equivariance, and furthermore showed that relaxed equivariant layers can be constructed as solutions to systems of equations.

Perhaps most closely related to our work is Xie & Smidt (2024). Inspired by the idea that symmetry breaking arises from missing information, Xie & Smidt (2024) defined an equivariant *symmetry breaking set* (SBS) $B(x)$ (for each input $x$) as a $G_x$-dependent equivariant set $B(x)$ on which $G_x$ acts freely. Elements $b \in B(x)$ can then be input to an equivariant function $f(x, b)$, breaking the symmetry of $x$ much like our $\tilde{g}$. Our work allows one to analyze SBS through a probabilistic lens: Proposition 5.1 implies that uniformly sampling from an SBS can represent any equivariant distribution, if one also adds $\epsilon \sim \text{Unif}(0, 1)$ to the input as $f(x, b, \epsilon)$. However, using an SBS requires detecting $G_x$ and its normalizer, while our framework naturally suggests implementation via existing canonicalization methods. Additionally, Xie & Smidt (2024) are largely concerned with identifying when an "ideal" $B(x)$, i.e. of minimal size $|B(x)| = |G_x|$, exists (which they argue should facilitate learning). Indeed, their chosen restriction that $B(x)$ depend only on $G_x$ means such an ideal SBS may not exist. In contrast, by allowing $\text{supp}(\tilde{g}) = G_x\tau(x)$ to depend on $x$ itself and not only $G_x$, we effectively always obtain an "ideal" set because $|G_x\tau(x)| = |G_x|$.

## 8 EXPERIMENTS

We evaluate SymPE empirically on three tasks: autoencoding graphs with EGNN (Section 8.1), graph generation with the DiGress diffusion process (Section 8.2), and predicting ground states of Ising models with G-CNNs (Section 8.3). In all cases, we find that SymPE outperforms baselines, both without symmetry-breaking and with other methods for symmetry-breaking.

Table 1: Cross-entropy loss and reconstruction error in graph autoencoding.

| Method | BCE | % Error | # Param. |
|--------|-----|---------|----------|
| No SB | 10.0 | 3.9 | 101,074 |
| Noise | 10.1 | 2.3 | 88,017 |
| Uniform | 5.7 | 1.3 | 88,890 |
| Laplacian | 5.6 | 0.98 | 88,885 |
| SymPE (ours) | **3.7** | **0.77** | 101,750 |

### 8.1 GRAPH AUTOENCODING / LINK PREDICTION

Autoencoders with symmetric latent spaces pose a problem for equivariant models. One example explored in Satorras et al. (2021) is autoencoding graphs using a node-wise latent space $\mathcal{Z} = \mathbb{R}^{n \times f}$, where $n$ is the number of nodes and $f$ is the feature dimension. This can alternatively be understood as a link prediction problem via node representations, for which equivariance is known to be problematic Srinivasan & Ribeiro (2020); Zhang et al. (2021). From an $S_n$-equivariant embedding in $\mathcal{Z}$, the graph is decoded equivariantly; the presence of an edge between nodes with latents $z_i$ and $z_j$ is a function of $\|z_i - z_j\|$. If $A$ is the adjacency matrix of the graph, its self-symmetries are $G_A = \{g \in S_n : gAg^T = A\}$. However, there may not even *exist* an embedding $z \in \mathcal{Z}$ such that $G_A = G_z$ (see Appendix H for details), which by Curie's Principle results in an "overly symmetric" embedding with $G_A \subsetneq G_z$ (Satorras et al., 2021, Figure 3) when *any* equivariant encoder is used.

We consider reconstructing Erdős-Rényi random graphs with edge probability $0.25$, using the data from Satorras et al. (2021) (which we compute, via `networkx`, contains at least 44% automorphic graphs) and their standard message-passing architecture as the encoder. To break symmetries using our method, we first compute learned scalar nodewise embeddings $x$ using one message-passing layer, and then obtain $\tilde{g}$ by sorting $y$, letting the sorting algorithm break ties. The symmetry breaking input $\tilde{v} = \tilde{g}v$ is obtained by correspondingly sorting a learned vector $v \in \mathbb{R}^n$, and is input to the encoder. As baselines, we consider no symmetry breaking[4] ("No SB"), randomly initialized node features ("Noise"), randomly sampling $\tilde{g}$ from $S_n$ ("Uniform"), and passing in spectral features directly ("Laplacian"). In Appendix H, we consider other baselines (such as breaking symmetries at the embedding level.) Breaking symmetries via our method achieves the lowest error (Table 1).

### 8.2 GRAPH GENERATION WITH DIFFUSION MODELS

We evaluate our framework on graph generation. Small graphs, which are of interest for molecular generation, are especially likely to have non-trivial automorphism groups (Godsil & Royle, 2001).

---

[4]To ensure a fair comparison to our method, we still compute $y$ and input it to the encoder.

We apply SymPE, our symmetry-breaking positional encoding, to discrete diffusion-based graph generation. We follow the setup and experimental protocol of a state-of-the-art method, DiGress (Vignac et al., 2023), in which a discrete diffusion process is applied on a graph's node features and adjacency matrix. A graph transformer then predicts back the denoised graph. Symmetry breaking is potentially crucial in this setting: since the denoising network is equivariant, if the diffusion process ever introduces a symmetry in the graph, it is impossible to denoise back to the original graph due to Curie's Principle. Vignac et al. (2023) note that a heuristic that adds spectral features to node features improves performance. These can indeed break symmetries, but in a less principled way than our method (see Appendix H), and at a computational cost scaling cubically in the number of vertices.

We evaluate combining our method with DiGress on the QM9 (Wu et al., 2017) and MOSES (Polykovskiy et al., 2020) datasets. For QM9, we consider the more challenging version in which hydrogen atoms appear explicitly in the graphs. We use sorting-based $S_n$ canonicalization with a GIN architecture (Xu et al., 2019) to sample $\tilde{g}$, following Kim et al. (2023). The symmetry-breaking positional encoding $\tilde{g}v_n$ is concatenated to node features, and $\tilde{g}v_e\tilde{g}^{-1}$ to the adjacency matrix, with learned $v_n \in \mathbb{R}^{n \times d}$ and $v_e \in \mathbb{R}^{n \times n \times d}$ (with $n$ set to the maximum graph size and $d = 8$). Further details on the experimental procedure are given in Appendix G.

Table 2: Evaluation metrics for molecular generation on QM9 with explicit hydrogens.

| Method | Valid↑ | Unique↑ | Atomic stability↑ | Mol. stability↑ | NLL |
|---|---|---|---|---|---|
| Dataset | 97.8 | 100 | 98.5 | 87.0 | - |
| ConGress (a variant of DiGress) | $86.7_{\pm1.8}$ | $\mathbf{98.4}_{\pm0.1}$ | $97.2_{\pm0.2}$ | $69.5_{\pm1.6}$ | - |
| DiGress (with Laplacian) | $95.4_{\pm1.1}$ | $97.6_{\pm0.4}$ | $98.1_{\pm0.3}$ | $79.8_{\pm5.6}$ | 129.7 |
| DiGress + SymPE (ours) | $\mathbf{96.1}$ | 97.5 | $\mathbf{98.6}$ | $\mathbf{82.5}$ | $\mathbf{30.3}$ |
| DiGress + noise | 90.7 | 97.6 | 97.8 | 73.1 | 126.5 |
| DiGress + SymPE (nodes only) | 96.2 | 97.4 | 98.4 | 83.9 | 128.8 |

Results for QM9 are shown in Table 2 and results for MOSES in Table 4 (Appendix G). Our method leads to a large improvement in negative log-likelihood (NLL) compared to the original DiGress. Note that the NLL captures the ability of the model to learn the right distribution, but not necessarily chemical validity of the generated samples. Another factor to consider is that for the other metrics, the baseline values are already close to the dataset values, making them more difficult to improve. We perform ablation studies with alternative methods to break symmetry. First, we consider breaking symmetry by concatenating noise sampled from a standard normal distribution ("DiGress + noise"). This does not lead to a similar increase in performance, which we hypothesize is because the model learns to ignore the uninformative noise. We also consider only using SymPE on the node features, and not on the adjacency matrix ("DiGress + SymPE (nodes only)"). This also does not have a significant effect on the likelihood, showing that breaking symmetry directly on the adjacency matrix fed to the graph transformer is crucial.

## 8.3 Predicting ground-states of Ising models

Spin systems are prototypical examples of physical systems that exhibit spontaneous symmetry breaking. Here, we consider unsupervised training of neural networks to obtain ground-states (states of minimal energy) of the Ising model given Hamiltonian parameters. The Ising model is an idealization of a magnetic system of spins, with the Hamiltonian describing its energy. Identifying low-energy configurations of spin systems is an important problem in statistical physics (Hu et al., 2017; Carrasquilla & Melko, 2017) and has applications to the graph max-cut problem (Fu & Anderson, 1986), yet brute force optimization scales exponentially in system size. Monte-Carlo simulation is possible, but does not benefit from the generalization of neural networks across Hamiltonians.

Formally, given a set of lattice sites $\Lambda$, a spin configuration $\sigma \in \{-1, 1\}^{\Lambda}$ specifies a binary value for each site. The Hamiltonian $H : \{-1, 1\}^{\Lambda} \rightarrow \mathbb{R}$ assigns an energy to each configuration $\sigma$. In our experiments, we consider the anisotropic Ising model under an external field $h$ on a square periodic lattice $\Lambda$, given by $H_{J,h}(\sigma) = -\sum_{i,j} J_{ij}^x \sigma_i \sigma_j - \sum_{i,j} J_{ij}^y \sigma_i \sigma_j - \sum_i h_i \sigma_i$. Here, $J_{ij}^x \neq 0$ only if $i$ and $j$ are horizontal neighbors in $\Lambda$, and similarly for $J^y$. The task has symmetry under the automorphism group of the square grid, $G = p4m$ (see Appendix F for details on the group action). That is, for any $g \in G$, we have $H_{gJ,gh}(g \cdot \sigma) = H_{J,h}(\sigma)$, so the energy is unchanged if both the

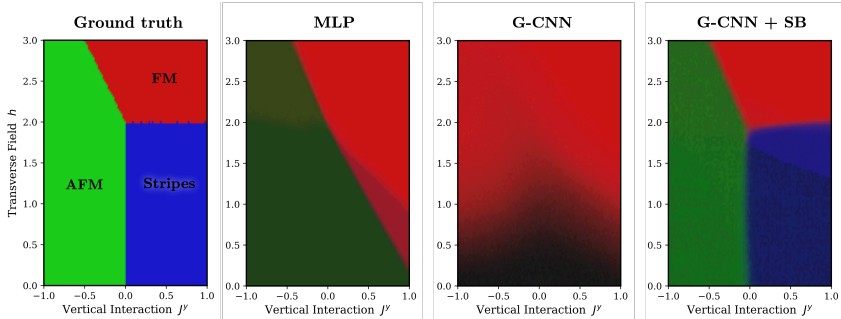

Figure 5: Phase diagrams predicted by the different methods. For each configuration predicted by the neural network on a test set Hamiltonian, we compute the values of the three order parameters: the ferromagnetic phase (red), the antiferromagnetic phase (green), and the stripes phase (blue). Brighter colors are associated with larger values of the order parameter, and black to the disordered phase.

Hamiltonian parameters and the spin configuration are transformed in the same way. The Hamiltonian parameters themselves have a self-symmetry to reflection and translations, which corresponds to the stabilizer group $G_{J,h} = pmm \subset G$. Ground-states are configurations of minimum energy, or non-zero probability, of the distribution $\mathbb{P}(\sigma|J,h) \propto \exp(-H_{J,h}(\sigma)/T)$ in the $T \to 0$ limit. While $\mathbb{P}(\sigma|J,h)$ is equivariant to $G$, ground states may break the $pmm$ self-symmetries of $(J,h)$.

We train a $p4m$-equivariant G-CNN combined with SymPE to take Hamiltonian parameters as input, and return a spin configuration as output. Training is done by directly using the Hamiltonian as the loss function and minimizing energy. We build a training and test set by sampling diverse Hamiltonian parameters $J$ and $h$ from a given distribution. To evaluates the ability to generalize to transformed data, we additionally build an out-of-distribution (OOD) test set for which Hamiltonian parameters are randomly rotated by $90°$. See Appendix F for details on the experimental setup.

We compare our method to a vanilla G-CNN, a non-equivariant MLP trained with data augmentation sampled from $p4m$, adding noise $\epsilon \sim \mathcal{N}(0,1)$ to the input of the G-CNN to break symmetry, and the relaxed group convolutions proposed by Wang et al. (2024). We use the average energy as an evaluation metric. The proposed method achieves significantly lower energy than baselines on the OOD test set, and is only slightly worse than the relaxed convolutions method on the in-distribution test set. (Table 3, Appendix F). A more in-depth understanding of the baselines's shortcomings can be obtained from the predicted phase diagrams (Fig. 5). Spin systems show phase transitions depending on Hamiltonian parameters, similar to molecular systems (see Appendix F). For the considered Ising models, there are three possible phases: ferromagnetism, antiferromagnetism, and stripes order. The order parameters give a quantitative description of the predicted phases given Hamiltonian parameters. We see that the G-CNN is not able to predict antiferromagnetic and stripes phases, which break symmetries. The MLP also does not recover the correct phase diagram.

# 9 CONCLUSION

By considering symmetry-breaking from a probabilistic perspective, we derive a representation of any equivariant distribution in terms of a deterministic equivariant function of the input, and a symmetry-breaking sample from the inversion kernel. Motivated by this theoretical result, we introduce a flexible method for breaking symmetries in existing equivariant architectures, by concatenating a random symmetry-breaking positional encoding (SymPE). We show that our method is a special case (of lowest entropy) within the general class of equivariant noise-injection methods, which we prove are able to represent equivariant distributions and enjoy guaranteed generalization benefits. We observe in experiments that SymPE outperforms baselines, both with and without symmetry-breaking, on graph autoencoding, graph generation, and Ising model ground-state prediction.

One limitation of SymPE is that it requires access to a canonicalization method, which may be less readily available for uncommon groups, as well as a way to randomize its outputs. Future work remains to sample $\tilde{g}$ in practice for general (including infinite) groups, such as: (1) testing our proposal of using energy-based modeling in a generic setting, (2) sampling from low-entropy equivariant distributions when the inversion kernel may be hard to sample from exactly, and (3) further exploring the tradeoff between structured symmetry breaking and generic noise injection. It also remains to address *partial* symmetry breaking (when the output breaks some, but not all, symmetries of the input), as treated for example by Xie & Smidt (2024).

ACKNOWLEDGMENTS

This research was enabled in part by computational resources provided by Mila, Compute Canada, and MIT Supercloud. S.-O. K. is supported by IVADO and the DeepMind Scholarship. H.L. is supported by the Fannie and John Hertz Foundation. V.P. is supported by the Gatsby Charitable Foundation. The authors are grateful to Siamak Ravanbakhsh, Daniel Levy, Tara Akhound-Sadegh, Vitória Barin Pacela, Bonaventure F. P. Dossou, Clément Vignac, Matthew Morris, Tess Smidt, YuQing Xie, Elyssa Hofgard, Nadav Dym, Jonathan Siegel and Peter Orbanz for insightful discussions.

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

APPENDICES

## A   PROOFS

Our results hold for any $G$ acting *properly* on $\mathcal{X}$, which includes compact groups and actions on Riemannian manifolds by isometries. For more details, including the relationship of proper actions to needed measurability conditions, we refer the reader to Chiu & Bloem-Reddy (2023).

Theorem 3.4 is a consequence of the following result, which says one can represent any equivariant conditional distribution by randomly canonicalizing some function.

**Theorem A.1** (Randomized canonicalization). *$\mathbb{P}\left(Y|X\right)$ is equivariant if and only if*

$$Y \stackrel{a.s.}{=} \tilde{g}\phi(\gamma(X),\epsilon) \stackrel{a.s.}{=} \tilde{g}\phi(\tilde{g}^{-1}X,\epsilon) \tag{5}$$

*for some function $\phi : \mathcal{X} \times (0,1) \to \mathcal{Y}$, independent noise $\epsilon \sim \mathrm{Unif}(0,1)$, and $\tilde{g}|X$ distributed according to the inversion kernel for some orbit representative map $\gamma$.*

The proof (Appendix A.1) follows closely that of Bloem-Reddy and Teh, which bootstraps *deterministic* symmetry from a canonicalizer to that of $\mathbb{P}\left(Y|X\right)$. We analogously bootstrap the *distributional* symmetry of $\mathbb{P}\left(\tilde{g}|X\right)$ to that of a model $\tilde{g}\phi(\tilde{g}^{-1}X,\epsilon)$ for $Y$, where $\phi$ is an arbitrary function.

In addition to Theorem 3.4, the above allows us to derive a representation in terms of *relaxed equivariant* functions as defined by Kaba & Ravanbakhsh (2023).

**Definition A.2.** A function $f : \mathcal{X} \to \mathcal{Y}$ is *relaxed equivariant* if for any $x \in \mathcal{X}$ and $g \in G$, we have $hf(x) = f(gx)$ for some $h \in gG_x$.

On $x$ with trivial stabilizer, a relaxed equivariant function behaves like a usual equivariant function. On points with non-trivial self-symmetry, relaxed equivariance only enforces symmetry of $f$ "up to elements in the stabilizer," so the output need not have any self-symmetry.

**Corollary A.3** (Relaxed equivariance). $\mathbb{P}\,(Y|X)$ *is equivariant if and only if*

$$Y \stackrel{a.s.}{=} g_X f(X, \epsilon) \tag{6}$$

*for some $f : \mathcal{X} \times (0,1) \to \mathcal{Y}$ relaxed equivariant in its first input, with $g_X|X \sim \mathrm{Unif}(G_X)$, and $\epsilon \sim \mathrm{Unif}(0,1)$ independent of $X$ and $g_X$.*

The proof is in Appendix A.3.

## A.1 PROOF OF THEOREM A.1

*Proof.* $\mathbb{P}\,(Y|X)$ is equivariant if and only if (Chiu & Bloem-Reddy, 2023, Theorem 3)

$$(\tilde{g}, X) \perp \tilde{g}^{-1}Y \mid \gamma(X) \tag{7}$$

where $\tilde{g}|X$ is distributed according to the inversion kernel associated to $\gamma$. By conditional noise outsourcing (Kallenberg, 2021, Proposition 8.20) this is equivalent to there existing a measurable function $\phi : \mathcal{X} \times (0,1) \to \mathcal{Y}$ such that

$$\tilde{g}^{-1}Y \stackrel{a.s.}{=} \phi(\gamma(X), \epsilon) \tag{8}$$

where $\epsilon \sim \mathrm{Unif}(0,1)$ is independent of $(\tilde{g}, X)$. Rearranging and noting $\tilde{g}\gamma(X) \stackrel{a.s.}{=} X$ gives the result. $\square$

## A.2 PROOF OF THEOREM 3.4

*Proof.* The forward implication is clear, letting $f(x, g, \epsilon) = g\phi(g^{-1}x, \epsilon)$ with $\phi$ given in the previous theorem. On the other hand consider a function $f : \mathcal{X} \times G \times (0,1)$ such that $hf(x, g, \epsilon) = f(hx, hg, \epsilon)$, and suppose $Y \stackrel{a.s.}{=} f(X, \tilde{g}, \epsilon)$. Write $\tilde{g}_x$ for a random variable sampled from the inversion kernel conditioned on $X = x$, independently from $\epsilon \sim \mathrm{Unif}(0,1)$. For any $x \in \mathcal{X}$ and $h \in G$ we have

$$\mathbb{P}\,(hY \in B|hX = x) = \mathbb{P}\,(hY \in B|X = h^{-1}x) = \mathbb{P}\,(hf(h^{-1}x, \tilde{g}_{h^{-1}x}, \epsilon) \in B). \tag{9}$$

By the equivariance of the inversion kernel, $\tilde{g}_{h^{-1}x} \stackrel{d}{=} h^{-1}\tilde{g}_x$. Applying first this fact and then the equivariance of $f$ we obtain

$$\mathbb{P}\,(hY \in B|hX = x) = \mathbb{P}\,(hf(h^{-1}x, h^{-1}\tilde{g}_x, \epsilon) \in B) = \mathbb{P}\,(f(x, \tilde{g}_x, \epsilon) \in B) = \mathbb{P}\,(Y \in B|X = x). \tag{10}$$

Note that for this direction, we only needed the equivariance of the distribution $\mathbb{P}\,(\tilde{g}|X)$, not its restriction to a specific coset. $\square$

## A.3 PROOF OF COROLLARY A.3

*Proof.* Given $\phi$ from the previous theorem, let $f$ be a relaxed equivariant function such that $f(\gamma(x), \epsilon) = \phi(\gamma(x), \epsilon)$ (which we can always do).

$$Y \stackrel{a.s.}{=} \tilde{g}\phi(\gamma(X), \epsilon) = \tilde{g}f(\gamma(X), \epsilon) = \tilde{g}h^{-1}f(X, \epsilon) \tag{11}$$

for some $h \in G$ (depending on the construction of $f$) such that $h\gamma(X) = X$. But $\tilde{g}|X \sim \mathrm{Unif}(hG_{\gamma(X)})$ and $hG_\gamma(X)h^{-1} = G_X$, so $\tilde{g}h^{-1}|X \sim \mathrm{Unif}(G_X)$.

Suppose on the other hand $Y \overset{a.s.}{=} g_X f(X, \epsilon)$ for a relaxed equivariant $f$. Write $g_x$ for a uniform random element of $G_x$ independent of $\epsilon \sim \mathrm{Unif}(0,1)$. We then have

$$\mathbb{P}(hY \in B | hX = x) = \mathbb{P}(hY \in B | X = h^{-1}x) = \mathbb{P}(hg_{h^{-1}x} f(h^{-1}x, \epsilon) \in B). \quad (12)$$

Since $g_{h^{-1}x} \overset{d}{=} h^{-1} g_x h$,

$$\mathbb{P}(hY \in B | X = h^{-1}x) = \mathbb{P}(g_x h f(h^{-1}x, \epsilon) \in B). \quad (13)$$

By the relaxed equivariance of $f$, for some $k^{-1} \in h^{-1} G_x$, the above is equal to

$$\mathbb{P}(g_x h k^{-1} f(x, \epsilon) \in B). \quad (14)$$

Noting that $hk^{-1} \in G_x$, we have $g_x \overset{d}{=} g_x h k^{-1}$ and thus

$$\mathbb{P}(hY \in B | hX = x) = \mathbb{P}(Y \in B | X = x). \quad (15)$$

(The invariance of the distribution of $g_X | X$ is used here, though what is strictly needed is equality in distribution of $h g_{h^{-1}x} k^{-1}$ and $g_x$.) $\qquad \square$

A.4 PROOF OF PROPOSITION 5.1

We decompose the forward and reverse implications in the following four statements:

1. If the action of $G$ on $\mathcal{Z}$ is free except for a measure zero subset, then equivariance of $\mathbb{P}(Y|X)$ implies that $Y \overset{a.s.}{=} f(X, Z, \epsilon)$ for an appropriate $f$ and $\epsilon$. Note that equivariance of $Z|X$ is not needed here.

2. If the action of $G$ on $\mathcal{Z}$ is free except for a measure zero subset and $\mathbb{P}(Z|X)$ is equivariant, then $Y \overset{a.s.}{=} f(X, Z, \epsilon)$ implies equivariance of $\mathbb{P}(Y|X)$.

3. If the action of $G$ on $\mathcal{X} \times \mathcal{Z}$ is *not* free except for a measure zero subset, then $Y \overset{a.s.}{=} f(X, Z, \epsilon)$ is *not* equivalent to equivariance of $\mathbb{P}(Y|X)$

4. If $\mathbb{P}(Z|X)$ is *not* equivariant, then $Y \overset{a.s.}{=} f(X, Z, \epsilon)$ is *not* equivalent to equivariance of $\mathbb{P}(Y|X)$

*Proof of 1.* Let $\tilde{g}$ be a group element distributed according to the inversion kernel. Reusing arguments from our main results we show there exists a map $t : \mathcal{X} \times \mathcal{Z} \times (0,1) \to G$ equivariant in the first two inputs such that $\tilde{g} \overset{a.s.}{=} t(X, Z, \eta)$ where $\eta \sim \mathrm{Unif}(0,1)$ is random noise independent of $X$ and $Z$. By Corollary 3.4 $Y \overset{a.s.}{=} f_0(X, t(X, Z, \eta), \epsilon_0)$ for a jointly equivariant $f_0$. The unit interval and unit square both being standard probability spaces, there exists a measure preserving bijection $\epsilon \leftrightarrow (\eta, \epsilon_0)$. We thus define $f(X, Z, \epsilon) = f_0(X, t(X, Z, \eta), \epsilon_0)$.

To show the existence of $t$, one repeats the proof of Theorem A.1 and Corollary 3.4, but applying Chiu & Bloem-Reddy (2023, Theorem 3) in the special case of an essentially free action. In particular, since $(X, Z)$ has trivial stabilizer almost surely,

$$\tilde{g}^{-1} Y \perp (X, Z) \mid \gamma(X, Z), \quad (16)$$

where $\gamma(X, Z) = (\gamma(X), Z)$ (with a slight abuse of notation). The rest of the proof is identical to that of the forward direction of Corollary 3.4. $\qquad \square$

*Proof of 2.* The proof is identical to that of the reverse direction of Corollary 3.4. $\qquad \square$

*Proof of 3.* Suppose $G$ does not act on $\mathcal{Z}$ essentially freely. Suppose $\mathbb{P}(Y|X)$ is equivariant and there is symmetry breaking with non-zero probability, i.e. $\mathbb{P}(G_{X,Z} \not\subseteq G_Y) > 0$. (Such examples are not hard to construct.) Suppose for the sake of contraction that also $Y \overset{a.s.}{=} f(X, Z, \epsilon)$ with some $f$ and $\epsilon$ as usual. Curie's Principle gives the contradiction: $G_{X,Z} \subseteq G_Y$ at any realization of $\epsilon$. $\qquad \square$

*Proof of 4.* We consider non-equivariant $\mathbb{P}(Z|X)$; concretely, suppose $Z$ is constant $z \in \mathcal{Z}$, and that there exists $g \in G$ such that $gz \neq z$. Let $f : \mathcal{X} \times \mathcal{Z} \times (0,1) \to \mathcal{Y}$ be the jointly equivariant function $f(x, z, \epsilon) = x \oplus z$ (so $\mathcal{Y} = \mathcal{X} \times \mathcal{Z}$). Then if $Y \stackrel{a.s.}{=} f(X, z, \epsilon)$,

$$\mathbb{P}(gY = y|gX = x) = \mathbb{P}\left(gf(X, z, \epsilon) = y|X = g^{-1}x\right) = \mathbb{1}\{x \oplus gz = y\} \tag{17}$$

$$\neq \mathbb{1}\{x \oplus z = y\} = \mathbb{P}(Y = y|X = x). \tag{18}$$

That is, $\mathbb{P}(Y|X)$ is not equivariant. $\qquad\square$

## A.5 PROOF OF THEOREM 6.1

The proof of Theorem 6.1 resembles that of Lemma 3.12 in Elesedy (2023). The latter applies the orthogonal decomposition of Elesedy & Zaidi (2021) in combination with the result of Bloem-Reddy & Teh (2020). Analogously, we combine the orthogonality arguments with our Proposition 5.1.

*Proof.* The assumption that $G$ acts essentially freely on $\mathcal{Z}$ and the equivariance of $\mathbb{P}(Y|X)$, by part 1 of our proof of Proposition 5.1 (Appendix A.4), means that there exists $f : \mathcal{X} \times \mathcal{Z} \times (0,1) \to \mathcal{Y}$ jointly equivariant in its first two coordinates such that $Y \stackrel{a.s.}{=} f^*(X, Z, \epsilon)$ for independent noise $\epsilon \sim \text{Unif}(0,1)$. Defining for each $\epsilon$ the function $f_\epsilon^* : (x, z) \mapsto f^*(x, z, \epsilon)$, the finiteness of $\mathbb{E}[\|Y\|^2]$ implies $f_\epsilon \in L^2(\mathcal{X} \times \mathcal{Z}, \mathcal{Y}, \mathbb{P}(X, Z))$ for almost every $\epsilon$. Note next that the invariance of $\mathbb{P}(X)$ and equivariance of $\mathbb{P}(Z|X)$ imply the invariance of the joint distribution $\mathbb{P}(X, Z)$. We can therefore apply Lemma 1 of Elesedy & Zaidi (2021) to obtain decomposition $f = \bar{f} + f^\perp$ into equivariant and "anti-equivariant" components, orthogonal under the inner product $\langle f_1, f_2 \rangle_{\mathbb{P}(X,Z)} = \int \int \langle f_1(x, z), f_2(x, z) \rangle_{\mathcal{Y}} \mathbb{P}(dx, dz)$. The risk of $f$ may then be written as

$$\mathbb{E}[\|f - f_\epsilon^*\|_{\mathbb{P}(X,Z)}^2] = \mathbb{E}[\|\bar{f} - f_\epsilon^*\|_{\mathbb{P}(X,Z)}^2] + \|f^\perp\|_{\mathbb{P}(X,Z)}^2 = R(\bar{f}) + \|f^\perp\|_{\mathbb{P}(X,Z)}^2, \tag{19}$$

where we obtain the first expression by conditioning on $\epsilon$, and the equality follows from the orthogonality of $f^\perp$ and $f_\epsilon^*$. The theorem follows by subtracting $R(\bar{f})$ from both sides of the equation. $\quad\square$

## B GENERALIZATION BENEFITS OF EQUIVARIANT CONDITIONAL DISTRIBUTIONS

Here, we are interested in describing the generalization benefits of using equivariant conditional distributions, when the ground truth distribution $\mathbb{P}(Y|X)$ is equivariant. The general outline, which we will fill in below, follows the work of Elesedy & Zaidi (2021) and its straightforward generalization in Elesedy (2023). We similarly assume that $X$ has a $G$-invariant distribution, that $\mathbb{P}(Y|X)$ is equivariant, and the group is compact.[5] Then, considering a Hilbert space of conditional distributions (or rather, their unnormalized counterparts), one can show that the equivariant ones form a subspace. The generalization benefits of assuming equivariance can then be expressed in terms of the projection operator onto that subspace.

In order to follow the program above, we will treat conditional distributions as equivariant functions $f : \mathcal{X} \to \mathbb{P}(\mathcal{Y})$. Furthermore, in order to work in a Hilbert space, we restrict ourselves to distributions with a square integrable density with respect to some measure $dy$ on $\mathcal{Y}$. We assume the measure is invariant under $G$—the canonical example being $\mathcal{Y}$ a finite-dimensional Euclidean space with Lebesgue measure and $G$ acting orthogonally. Then, we define $\mathbb{P}(\mathcal{Y}) \subset L^2(\mathcal{Y})$ by

$$\mathbb{P}(\mathcal{Y}) = \left\{ \psi : \mathcal{Y} \to \mathbb{R} \ s.t. \ \psi(\cdot) \geq 0, \int_{y \in \mathcal{Y}} \psi(y) \, dy = 1, \int_{y \in \mathcal{Y}} \psi(y)^2 \, dy < \infty \right\} \tag{20}$$

We will treat densities $p(y|x)$ as members of the Hilbert space $\mathcal{H} = L^2(\mathcal{X}, L^2(\mathcal{Y}), \mathbb{P})$, where the data distribution $\mathbb{P}$ on $\mathcal{X}$ is $G$-invariant. The inner product between functions $f_1, f_2 \in \mathcal{H}$ is

$$\langle f_1, f_2 \rangle = \int_{\mathcal{X}} \langle f_1(x), f_2(x) \rangle \, \mathbb{P}(dx) = \int_{\mathcal{X}} \int_{\mathcal{Y}} f_1(y|x) f_2(y|x) \, dy \, \mathbb{P}(dx),$$

---

[5]Our previous results held in the more general case of proper group actions.

where for simplicity we use the notation $f(y|x) = (f(x))(y)$ for even those $f \in \mathcal{H}$ which are not probability densities.

$G$ acts on functions $\psi \in L^2(\mathcal{Y})$—and thus on $\mathbb{P}(\mathcal{Y})$—by

$$(g \cdot \psi)(y) = \psi(g^{-1}y).$$

By the assumption of the $G$-invariance of $dy$, the inner product on $L^2(\mathcal{Y})$ is invariant: $\langle \psi_1, \psi_2 \rangle = \langle g\psi_1, g\psi_2 \rangle$. Standard arguments from Elesedy (2023, Lemma 3.1) then show that the *Reynolds operator* $\mathcal{R} : \mathcal{H} \to \mathcal{H}$ given by[6]

$$(\mathcal{R}f)(x) = \int_G g^{-1} f(gx) \, \lambda(dg) \tag{21}$$

$$(\mathcal{R}f)(y|x) = \int_G f(gy|gx) \, \lambda(dg) \tag{22}$$

is in fact the orthogonal projection onto the subspace of equivariant functions—i.e. those $f \in \mathcal{H}$ such that $f(g^{-1}y|x) = f(y|gx)$. For the Reynolds operator to be useful to us, it remains to check that it sends normalized conditional densities $p(y|x)$ to normalized conditional densities (which will then be equivariant). But this is clear:

$$\int_{\mathcal{Y}} (\mathcal{R}p)(y|x) \, dy = \int_{\mathcal{Y}} \int_G p(gy|gx) \, \lambda(dg) \, dy = \int_G \int_{\mathcal{Y}} p(gy|gx) \, dy \, \lambda(dg) = 1, \tag{23}$$

where at the end we used the invariance of $dy$, and the fact that $p(y|x)$ and $\lambda$ are normalized.

We then consider risk under the $L^2(\mathcal{Y})$ loss. The *generalization gap* between two conditionals $p_1, p_2 \in \mathcal{H}$

$$\Delta(p_1, p_2) = R(p_1) - R(p_2) \tag{24}$$

where $R$ is the risk as measured against the ground truth conditional $p^*$,

$$R(p) = \int_{\mathcal{X}} ||p(x) - p^*(x)||^2_{L^2(Y)} \, \mathbb{P}(dx) = \int_{\mathcal{X}} \int_{\mathcal{Y}} (p(y|x) - p^*(y|x))^2 \, dy \, \mathbb{P}(dx). \tag{25}$$

We can rewrite that risk as

$$R(p) = ||p - p^*||^2_{\mathcal{H}} = ||p||^2_{\mathcal{H}} - 2\langle p, p^* \rangle_{\mathcal{H}} + ||p^*||^2_{\mathcal{H}}. \tag{26}$$

(We subsequently drop the subscripts for conciseness.) The generalization gap between an arbitrary $p \in \mathcal{H}$ and its equivariant projection $\bar{p} = \mathcal{R}p$ is then given in terms of the orthogonal component $p^\perp = p - \bar{p}$:

$$\Delta(p, \bar{p}) = ||p - p^*||^2 - ||\bar{p} - p^*||^2 \tag{27}$$

$$= ||\bar{p} + p^\perp - p^*||^2 - ||\bar{p} - p^*||^2 \tag{28}$$

$$= ||\bar{p} - p^*||^2 + ||p^\perp||^2 - ||\bar{p} - p^*||^2 = ||p^\perp||^2 \tag{29}$$

where one gets to the the last line by using the orthogonality of $p^\perp$ and $\bar{p}, p^* \in \mathcal{H}_G$.

## C    IMPLEMENTATION OF INVERSION KERNELS

We now provide more details on the implementation of inversion kernels. As noted in the main text, a general method is provided by the following optimization approach of Kaba et al. (2023). Given an energy function $E : \mathcal{X} \to \mathbb{R}$, there exists a canonicalization $\tau$ such that the set $\arg\min_{g \in G} E(g^{-1}x) = G_x\tau(x)$, so long as the energy function is "non-degenerate". By that, we mean that there is a unique minimizer $\gamma(x) = \arg\min_{x' \in Gx} E(x')$ for any orbit $Gx$. For different groups, we describe how an energy function $E$ can be parameterized such that the optimization is efficient. We also give more details on the non-degeneracy requirement, which is non-trivial to satisfy for some groups, but does not pose issues in practice.

---

[6]We use $\lambda$ to denote the (normalized) Haar measure on $G$.

**Discrete translation and rotation groups with equivariant neural networks**  As shown by Kaba et al. (2023), the energy function $E$ can in general be alternatively represented with an equivariant function. This can be formalized with the following proposition.

**Proposition C.1.** *Suppose* $s : G \times \mathcal{X} \to \mathbb{R}$ *is a jointly equivariant function such that* $s(g, hx) = s(h^{-1}g, x) \,\forall g, h \in G, x \in \mathcal{X}$. *Then, there is a function* $E : \mathcal{X} \to \mathbb{R}$, *such that* $s(g, x) = E(g^{-1}x)$.

*Proof.* To prove this, we show that $s(g, x)$ only depends on $g^{-1}x$. Consider any two pairs $(g, x)$ and $(g', x')$ such that $g^{-1}x = g'^{-1}x'$. Then, it must be that there is an $h \in G$ such that $g' = hg$ and $x' = hx$. If we then consider $s(g', x') = s(hg, hx)$, using the equivariance condition we obtain $s(g', x') = s(g, x)$. $\qquad\square$

By currying, we can also see the equivariant function $s$ as outputting a real number for each group element, e.g. $s : \mathcal{X} \to \mathbb{R}^G$. For finite groups, like discrete roto-translations groups, the function $s$ can be conveniently implemented using equivariant neural network architectures like CNNs (LeCun et al., 1995) and G-CNNs (Cohen & Welling, 2016). By averaging the output feature map over channels (but not over fibers), we can obtain a real number for each group element and take the $\arg\max$ to sample from the inversion kernel. With this parametrization, the non-degeneracy requirement is heuristically expected to be satisfied. This is because for it not to be satisfied, the weights of the neural network would have to "accidentally" result in identical outputs for different group elements.

**Symmetric group with sorting**  The above parametrization in terms of an equivariant function $s : \mathcal{X} \to \mathbb{R}^G$ is impractical for the symmetric group $S_n$, since its size grows combinatorially with input size $n$ (here, we assume $\mathcal{X}$ consists of sets of $n$ objects). For this group, there is however a simple and efficient way to canonicalize using sorting. This can also be seen as an optimization procedure as follows.

We first define the energy as $E(g^{-1}x) = f(g^{-1}x) \cdot \rho^T$, where $f : \mathcal{X} \to \mathbb{R}^n$ is an $S_n$-equivariant function that scores each element of the set and $\rho = [n, n-1, \dots, 1]^T$. From equivariance, we have $E(g^{-1}x) = g^{-1}f(x) \cdot \rho^T$. Then, as shown by e.g. Blondel et al. (2020), $\arg\min_{g \in G} E(g^{-1}x) = \arg\min_{g \in G} g^{-1}f(x) \cdot \rho^T = \operatorname{argsort} f(x)$. We can therefore simply take $f$ to be any $S_n$-equivariant neural network, and sort the outputs corresponding to each element of an input set or nodes in a graph. When the input has self-symmetries, multiple permutations will sort the input, so one is chosen randomly from that set.

Note that using this method for graphs will not allow us to sample from an inversion kernel in general, since the non-degeneracy of the corresponding energy function cannot be guaranteed. This problem is related to the difficulty of canonicalizing graphs (Babai & Luks, 1983; McKay & Piperno, 2014). However, luckily, the fact that we do not sample from an inversion kernel does not restrict our ability to represent arbitrary equivariant conditional distributions. Since we still sample $\tilde{g}$ from an equivariant conditional distribution, Proposition 5.1 ensures that the representational power of the method is preserved. However, recall the conjecture that sampling from an equivariant conditional distribution of minimal entropy is best for learning (as noted in the main body, as well as motivating the search for "ideal" SBSs in Xie & Smidt (2024)). From this perspective, the closer to a true/powerful graph canonicalization is used, the easier learning may be.

**Continuous groups**  For continuous groups, in principle the energy minimization can be performed with gradient-based methods. It would, however, be impractical to require optimization until convergence. If this is not the case, or if the energy is degenerate, we do not formally sample from an inversion kernel. However, we still sample $\tilde{g}$ from an equivariant conditional distribution, which is compatible with Proposition 5.1. Heuristically, stochastic gradient-descent with a small learning rate is similar to Langevin sampling of the distribution $\exp(-E(g^{-1}x)/T)$ where $T$ is related to the stochasticity of the optimization (Welling & Teh, 2011). Moreover, some existing canonicalization methods are easily adapted to sampling from equivariant conditional distributions. For example, if one canonicalizes a point cloud by defining a coordinate axis based on the center of mass and two points of maximal radius, one can randomly sample these points of maximal radius in the case of a tie (which arises under self-symmetry).

# D  BREAKING SYMMETRY IN DIFFERENT GROUPS

In this appendix, we show how to learn symmetry breaking biases $v \in V$ (such that $G_v$ is trivial) for different groups. For any group $G$, the first basic requirement is for the group to act faithfully on $V$. We then want to choose $V$ such that we can initialize $v \in V$ and obtain $G_v$ with probability 1 (with the assumption that $v$ is initialized by sampling from an absolutely continuous distribution).

## D.1  PERMUTATION GROUPS

For permutation groups such as $S_n$ and $p4m$ (the symmetry group of an image grid), the group admits a faithful representation that maps to permutation matrices acting on $\mathbb{R}^n$. We can therefore choose $V = \mathbb{R}^n$.

In this case, it suffices that all the elements of a symmetry breaking bias $v$ are different. This is captured by the following proposition.

**Proposition D.1.** *Let $G$ act faithfully by permutation on $\mathbb{R}^n$ and $v \in \mathbb{R}^n$ be such that $v_i \neq v_j$ for any $i \neq j$. Then $G_v$ is trivial. In addition, the set of $v$ not satisfying this condition is of measure zero with respect to the Lebesgue measure.*

*Proof.* The proof of the first part of the proposition is trivial. For any $g \in G$, $(gv)_i = v_{g^{-1}(i)}$. Since the group action is faithful, for any $g \in G$ except the identity, there is an $i \in [n]$ such that $g^{-1}(i) \neq i$. For this $i$, $v_{g^{-1}(i)} \neq v_i$, therefore $G_v$ is trivial.

The second part follows the same idea as Proposition 3 of Kaba & Ravanbakhsh (2023). Consider the hyperplanes in $\mathbb{R}^n$, defined as $H_{ij} = \{v \in \mathbb{R}^n \mid v_i = v_j\}$. Any $v$ not satisfying the condition is an element of $S = \cup_{i \neq j}^n H_{ij}$. A hyperplane $H_{ij}$ defines an $(n-1)$-dimensional space in $\mathbb{R}^n$. Any subspace of $\mathbb{R}^n$ of dimension strictly less than $n$ is of measure zero. The countable union of such subspaces $S$ is also of measure zero. $\square$

## D.2  SUBGROUPS OF THE GENERAL LINEAR GROUP

We also consider groups that admit faithful representations as subgroups of $GL(n)$, such as $O(n)$ for point clouds and atomic systems. In this case, we can choose the symmetry breaking bias to be $n$ linearly independent vectors, so that $V = \mathbb{R}^{n \times n}$.

**Proposition D.2.** *Let $G \subseteq GL(n)$ and $V = \mathbb{R}^{n \times n}$. Assume $G$ acts on $V$ as a product of faithful actions, e.g. $gv \mapsto [gv^1, \ldots, gv^n]$, where $v^i$ is the $i$-th column vector of $v$. If the vectors $[v^1, \ldots, v^n]$ are linearly independent, then $G_v$ is trivial. In addition, the set of $v$ such that this condition is not satisfied is of measure zero with respect to the Lebesgue measure.*

*Proof.* For the first part, we can identify the action of the group and matrix multiplication $gv$. For any $g \in G_v$, it must be that $gv = v$. Since the columns of $v$ are linearly independent, $v$ is invertible. We therefore have $gvv^{-1} = vv^{-1}$, which implies $g = I$.

For the second part, the idea is similar to the proof of Proposition D.1 above. If two columns of $v$ are not linearly independent, it implies that $v$ is element of a subspace $H_{ij} = \{v \in \mathbb{R}^{n \times n} \mid \exists a \in \mathbb{R} \text{ s.t. } v^i = av^j\}$ for some $i$ and $j$. This is a subspace of measure zero since it is of dimension $n^2 - n + 1$. The union of all such subspaces $S = \cup_{i \neq j}^n H_{ij}$ is also of measure zero since it is countable. $\square$

## D.3  REPRESENTATION OF JOINTLY EQUIVARIANT FUNCTIONS

Here we show more formally that the encoding of group elements using vectors allows to represent any jointly equivariant function.

**Proposition D.3.** *Let $f : \mathcal{X} \times G \to \mathcal{Y}$ be any function jointly equivariant in its arguments, i.e. $f(hx, hg) = hf(x, g) \ \forall \ h \in G$, and let $V$ be a vector space on which $G$ acts. If $v \in V$ has trivial stabilizer, then there exists an equivariant function $f_0 : \mathcal{X} \times [v] \to \mathcal{Y}$ such that $f(x, g) = f_0(x \oplus gv)$.*

*Proof.* Consider any jointly equivariant $f : \mathcal{X} \times G \to \mathcal{Y}$, and suppose $v \in V$ has trivial stabilizer. We may let $f_0(x \oplus u) = f(x, g)$ where $g$ is the unique group element such that $gv = u$. $\qquad\square$

## E  THE INVERSION KERNEL INJECTS MINIMAL NOISE

We argue that using $\tilde{g}|X$ in place of independent $Z \in \mathcal{Z}$ introduces the least amount of noise possible into the functional representation of Proposition 5.1. We will measure "amount of noise" by conditional entropy, assuming $G$ is finite for ease of exposition. $\mathcal{Z}$ is a disjoint union of orbits. Since entropy is additive, it is minimized if $Z$ is restricted to a single orbit, which (assuming $G$ acts freely) is isomorphic to $G$ itself. We thus consider $Z$ as a $G$-valued random variable.

First notice that if $Z$ is independent of $X$, then it must be uniform on $G$ to be equivariant. It is easy to show the uniform distribution maximizes entropy. To see inversion kernels minimize it, let $g \in G$ be such that $\mathbb{P}\left(Z = g | X = x\right) = p$. If $\mathbb{P}\left(Z|X\right)$ is equivariant, then for any $h \in G_x$ we have $\mathbb{P}\left(Z = hg | X = x\right) = p$. Thus the support of $\mathbb{P}\left(Z | X = x\right)$ is at least the size of $G_x$. Entropy will again be minimized if $\mathbb{P}\left(Z | X = x\right)$ indeed has a support of this size. This is exactly the case when $Z|X$ is in fact distributed like $\tilde{g}|X$, according to an inversion kernel.

## F  ISING MODEL EXPERIMENTAL DETAILS

### F.1  GROUND STATE OF THE ANISOTROPIC ISING MODEL

We present here an analytical derivation of the ground states of the anistropic Ising model. This is used to obtain the ground-truth values for the average energy in Table 3 and the ground-truth phase diagram in Fig. 5.

The general form of the Hamiltonian of a spin system with binary interactions is given by

$$H\left(\sigma\right) = -\sum_{i,j} J_{ij}\sigma_i\sigma_j - \sum_i h_i\sigma_i. \tag{30}$$

The family of anisotropic Ising models we consider is given by

$$H(\sigma) = -\sum_{\langle i,j\rangle_x} J^x\sigma_i\sigma_j - \sum_{\langle i,j\rangle_y} J^y\sigma_i\sigma_j - \sum_i h\sigma_j n \tag{31}$$

where $\sum_{\langle i,j\rangle_x}$ indicates that the lattice sites $i$ and $j$ are nearest neighbors in the $x$ direction. We also consider periodic boundary conditions. We can re-express the Hamiltonian using the variables $b^x_{ij}, b^y_{ij} = [\sigma_i, \sigma_j] \in \{[1,1], [1,-1], [-1,1], [-1,-1]\}$ taking values on the horizontal and vertical interaction edges instead of the lattice sites. The variable encodes the value of the spins $i$ and $j$ adjacent to the bond. For convenience, we will write $b_{ij}$ as a one-hot vector, with

$$\mathbf{b}^\alpha_{ij} = \begin{cases} [1,0,0,0] & \text{if } b^\alpha_{ij} = [1,1] \\ [0,1,0,0] & \text{if } b^\alpha_{ij} = [1,-1] \\ [0,0,1,0] & \text{if } b^\alpha_{ij} = [-1,1] \\ [0,0,0,1] & \text{if } b^\alpha_{ij} = [-1,-1] \end{cases} \tag{32}$$

In terms of these variables, the Hamiltonian becomes

$$H(\mathbf{b}) = -\sum_{i,j} \begin{bmatrix} J^x \\ -J^x \\ -J^x \\ J^x \end{bmatrix} \mathbf{b}^x_{ij} - \sum_{i,j} \begin{bmatrix} J^y \\ -J^y \\ -J^y \\ J^y \end{bmatrix} \mathbf{b}^y_{ij} - \sum_{i,j} \frac{1}{2}\begin{bmatrix} h \\ 0 \\ 0 \\ -h \end{bmatrix} (\mathbf{b}^y_{ij} + \mathbf{b}^x_{ij}) \tag{33}$$

For the change of variables to correspond to a valid spin configuration, we additionally need to satisfy the constraints $b_{ij,0} = b_{ij',0}$ and $b_{i'j,1} = b_{ij,1}$ for any $i, i', j, j'$. Any configuration of variables satisfying such constraint will be an element of the feasible set $\mathcal{B}$.

Finding the ground state therefore corresponds to the optimization problem $\arg\min_{\mathbf{b}\in\mathcal{B}} H(\mathbf{b})$.

Since the form of the Hamiltonian Eq. (33) is a simple sum of non-interacting terms, we first directly minimize to find the ground-states without considering the constraint. We will then verify that the minimum lies in the feasible set $\mathcal{B}$.

We first re-write the Hamiltonian in the following way

$$H(\mathbf{b}) = -\sum_{i,j} \begin{bmatrix} J^x + h/2 \\ -J^x \\ -J^x \\ J^x - h/2 \end{bmatrix} \mathbf{b}_{ij}^x - \sum_{i,j} \begin{bmatrix} J^y + h/2 \\ -J^y \\ -J^y \\ J^y - h/2 \end{bmatrix} \mathbf{b}_{ij}^y \qquad (34)$$

We can then minimize each term independently to obtain the solution

$$\mathbf{b}_{ij}^\alpha = \begin{cases} [1,0,0,0] & \text{if } J^\alpha \geq -\frac{h}{4}, h \geq 0 \\ [0,1,0,0] \text{ or } [0,0,1,0] & \text{if } J^\alpha \leq -\frac{1}{4}|h| \\ [0,0,0,1] & \text{if } J^\alpha \geq \frac{h}{4}, h \leq 0 \end{cases} \qquad (35)$$

where $\alpha \in \{x, y\}$. Without loss of generality we consider $h \geq 0$. This leaves us with four possibilities.

1. $J^x \geq -\frac{h}{4}, J^y \geq -\frac{h}{4}$: The ground states is given by $\mathbf{b}_{ij}^\alpha = [1,0,0,0]$, which is in $\mathcal{B}$ and corresponds to the ferromagnetic (FM) phase (see Fig. 6a).

2. $J^x \leq -\frac{h}{4}, J^y \leq -\frac{h}{4}$: Feasible ground states consists of alternating between $\mathbf{b}_{ij}^\alpha = [0,1,0,0]$ and $\mathbf{b}_{ij}^\alpha = [0,0,1,0]$. This is indeed in $\mathcal{B}$ and corresponds to the antiferromagnetic (AFM) phase (see Fig. 6b).

3. $J^x \geq -\frac{h}{4}, J^y \leq -\frac{h}{4}$: Feasible ground states consists of having $\mathbf{b}_{ij}^x = [1,0,0,0]$ and alternating between $\mathbf{b}_{ij}^y = [0,1,0,0]$ and $\mathbf{b}_{ij}^y = [0,0,1,0]$. This is indeed in $\mathcal{B}$ and corresponds to the $y$ stripes phase ($S^y$) (see Fig. 6c).

4. $J^x \leq -\frac{h}{4}, J^y \geq -\frac{h}{4}$: This is the same as above but with stripes in the $x$ direction.

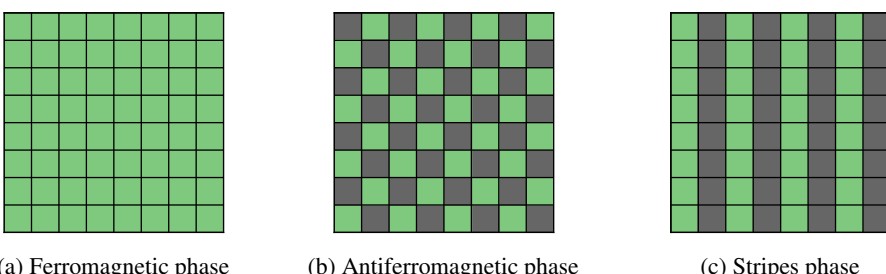

(a) Ferromagnetic phase      (b) Antiferromagnetic phase      (c) Stripes phase

Figure 6: Illustration of the different ground-states of the anisotropic Ising model

We can see that the different ground-states are associated with specific types of *orders*. The type of order of an arbitrary spin configuration $\sigma$, which is in a sense its closeness to the respective ground-states, can be quantified via *order parameters* (Beekman et al., 2019).

The order parameters are given by

$$O_{\text{FM}} = \frac{1}{N} \sum_i \sigma_i \qquad O_{\text{AFM}} = \frac{1}{N} \sum_i (-1)^{i^x + i^y} \sigma_i \qquad O_{S^y} = \frac{1}{N} \sum_i (-1)^{i^y} \sigma_i \qquad (36)$$

where $i^x$ and $i^y$ are respectively the $x$ and $y$ positions of a spin $\sigma_i$. The order parameters take the value 1 if and only if $\sigma$ is the associated ground-state. In addition, $O_{\text{FM}} + O_{\text{AFM}} + O_{S^y} \leq 1$, which means that the orders are mutually exclusive. This allows us to draw meaningful phase diagrams like the ones in Fig. 5.

### F.2 HAMILTONIAN ENCODING

For the encoding of the Hamiltonian interaction parameters as input to neural networks, a graph representation would be possible, with interaction parameters $J_{ij}$ encoded as edge attributes and external field values as node attributes (see Fig. 7a). However, this choice would make it much more challenging to use the $p4m$ symmetry of the Hamiltonian. We therefore leverage the structure of the

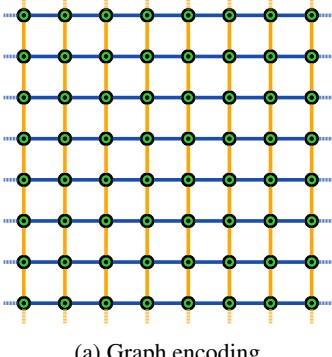
(a) Graph encoding

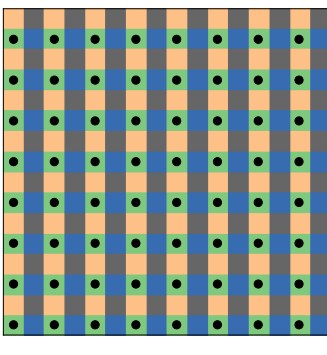
(b) Image encoding

Figure 7: Data structures for the encoding of the Hamiltonian interaction parameters. $J^x$ is represented in blue, $J^y$ in yellow and the external field in green. The lattice sites of the spins are represented with black dots. In the image, gray pixels corresponding to the "holes" between edges, are set to 0.

lattice and adopt an image representation (see Fig. 7b). We can then conveniently use G-CNNs and MLPs as prediction networks.

We choose to set the size of the spin grid to $64 \times 64$, which corresponds to images of size $128 \times 128$. We also encode the interaction parameters $J$ and the transverse field $h$ across two different channels. The dimension of the inputs to the models is therefore $[\text{batch}, 2, 128, 128]$

In intermediary activations of the neural network, we always preserve the size of the image. At the output of the network, we obtain a spin configuration by indexing the image over pixels corresponding to lattice sites (with black dots on figure Fig. 7b). The energy is then computed over the lattice sites.

### F.3 GROUP ACTION

We explicit here the action of the group $p4m$ (the symmetry group of the square lattice) on the Hamiltonian and on spin configurations.

To make the action clearer we unpack lattices indices variable in terms of horizontal and vertical components as $i \equiv (i^x, i^y)$. For $g \in p4m$, the action on a spin configuration is given by $g\sigma_{(i^x,i^y)} = \sigma_{(g^{-1}i^x, g^{-1}i^y)}$. The action on the indices permutes them.

The action on the Hamiltonian parameters is similarly given by $gh_{(i^x,i^y)} = h_{(g^{-1}i^x, g^{-1}i^y)}$ and $gJ_{(i^x,i^y),(i^x,i^y)} = J_{(g^{-1}i^x,g^{-1}i^y),(g^{-1}j^x,g^{-1}j^y)}$. In practice, action by the group on the image encoding of the Hamiltonian is simply given by performing the transformation on the image (Fig. 7b).

For any Hamiltonian, acting on both the parameters and the spin configuration preserves the energy:

$$
\begin{aligned}
H_{gJ,gh}(g\sigma) = &-\sum_{i,j} J_{(g^{-1}i^x,g^{-1}i^y),(g^{-1}j^x,g^{-1}j^y)} \sigma_{(g^{-1}i^x,g^{-1}i^y)} \sigma_{(g^{-1}j^x,g^{-1}j^y)} \\
&-\sum_i h_{(g^{-1}i^x,g^{-1}i^y)} \sigma_{(g^{-1}i^x,g^{-1}i^y)}.
\end{aligned}
\tag{37}
$$

which we see equals $H_{J,h}(\sigma)$ due to the sum over $i$ and $j$.

For the anisotropic Ising model we consider, the Hamiltonian parameters themselves are self-symmetric under the subgroup $pmm$, which includes translations, reflections and 2-fold rotations (but not 4-fold). In other words for any $g' \in pmm$, $g'J = J$ and $g'h = h$. This is easily seen from Fig. 7b, the image encoding exhibits a wallpaper pattern with symmetry $pmm$.

### F.4 TRAINING SETUP

Unsupervised training is performed by having the neural network $\phi : \mathcal{J} \to [0,1]^\Lambda$ output the probability that each spin is up by applying a softmax on the last layer; symmetry breaking elements are sampled through a canonicalization given by a G-CNN as in Kaba et al. (2023). When there is a

tie in the canonicalization, an element of the argmax set is chosen randomly. We use the EquiAdapt library (Mondal et al., 2023) to implement the canonicalization. While training, we then compute the expectation value of the spin at each site and treat this as the configuration, using the Hamiltonian as a loss function. At evaluation, we sample a spin value for each site using the probability output from the neural network.

We define a training set of Hamiltonian parameters as $J^x = -1$ and sampling $J^y \sim \mathrm{Unif}\,(-3, 3)$, $h \sim \mathrm{Unif}\,(0, 2)$, with parameters constant over the lattice. The training set is of size $1024$. For the validation we also use $1024$ samples. We consider two test sets: one in-distribution (ID) test set of $10,000$ regularly sampled Hamiltonian parameter values in the same range. We also consider an out-of-distribution (OOD) test set, which is the ID test set, augmented with rotations of the Hamiltonians parameters $(J, h)$. For each test set example, with 0.5 probability we act with the group element corresponding to $90°$ degree rotation. Note that it is not necessary to consider other augmentations in $p4m$ because they will either be self-symmetries the Hamiltonian parameters (be elements of the coset of $eG_{(J,h)}$) or will act in the same way as a $90°$ degree rotation (be elements of the coset of $gG_{(J,h)}$, with $g = 90°$).

## F.5 ENERGY RESULTS

We report energy results in Table 3. We see that for the ID test, the relaxed group convolutions method is slightly better than SymPE. However, for the OOD test set, SymPE becomes significantly better. This is due to the fact that SymPE retains an equivariance inductive bias compared to the relaxed convolutions methods which is not equivariant. Note that even for SymPE, the vanilla G-CNN and the G-CNN+noise, the results for the OOD test set are slightly worst even if the methods model equivariant conditional distributions. This can be attributed to border effects. For the non-equivariant MLP and the relaxed group convolutions, the decrease in performance is much more significant.

Table 3: Test set energies of predicted configurations

| Method | Energy (ID) | Energy (OOD) | Parameters | Forward time (s) |
|---|---|---|---|---|
| Random configurations | 0.0 | 0.0 | - | - |
| MLP | -1.22 | -0.93 | 3.2M | $8 \times 10^{-3}$ |
| MLP + aug. | -1.24 | -1.24 | 3.2M | $8 \times 10^{-3}$ |
| G-CNN | -0.69 | -0.69 | 397K | 0.47 |
| G-CNN + noise | -1.28 | -1.28 | 399K | 0.49 |
| Relaxed group convolution | **-1.49** | -1.42 | 463K | 0.47 |
| G-CNN + SymPE (Ours) | -1.46 | **-1.47** | 468K | 0.52 |
| Ground truth | -1.60 | -1.60 | - | - |

We also compare the parameter count and forward time (on Nvidia Quadro RTX 8000 GPUs with of the different models. We see that the computational overhead of SymPE is small compared to the vanilla G-CNN.

## G GRAPH DIFFUSION EXPERIMENTAL DETAILS

Our experimental setup follows exactly that of the original DiGress model, as described in Vignac et al. (2023), with the same discrete denoising diffusion process and graph transformer architecture.

We used the QM9 dataset of small molecules, incorporating explicit hydrogen atoms, for evaluating the validity and uniqueness of generated molecular graphs along with the negative log-likelihood of the test set. For the MOSES dataset we report the negative log-likelihood on the withheld test set, not on the scaffold test set. The graphs were preprocessed similarly to the standard setup in DiGress, where node features represent atom types and edge features represent bond types. We incorporated spectral features into the network to improve expressivity, following the methodology outlined in the original paper. The positional encodings introduced by SymPE were concatenated to node and edge features during the diffusion process.

In the Table 4, we report the negative log-likelihood results for MOSES.

Table 4: Negative log-likelihood for molecular generation on MOSES

| Method | NLL | Parameters | Training time (h) |
|---|---|---|---|
| DiGress | 65.9 | 16.2M | 28.12 |
| DiGress + SymPE (ours) | 30.4 | 16.3M | 34.61 |

## H  GRAPH AUTOENCODER EXPERIMENTAL DETAILS

As discussed in the main body, autoencoders with "very symmetric" latent spaces pose a problem for equivariant models. In the graph auto-encoding setup of Satorras et al. (2021), self-symmetry of a graph with adjacency matrix $A$ arises in the form of its automorphism group, $\{g \in S_n : gAg^T = A\}$. For example, the square graph $A$ in Figure 1 has $C_4$ automorphism group. When using a permutation-equivariant encoder $e$, $e(A)$ must therefore also have at least $C_4$ symmetry. Even though the goal of the permutation-equivariant decoder $d$ is to satisfy $d(e(A)) = A$, which would not seem to explicitly require symmetry-breaking, there is a subtle problem: the choice of latent space $\mathcal{Z}$. Precisely, $\mathcal{Z}$ does not contain any node-wise featurization with *precisely* $C_4$-symmetry: if $\{z_1, \ldots, z_4\} \in \mathcal{Z}$ have $C_4$-symmetry, this implies that $z_1 = \cdots = z_4$.[7] More abstractly, there exist graphs $A$, with stabilizer (automorphism group) $G_A$, such that there is no $Z \in \mathcal{Z}$ with $G_Z = G_A$; only $G_A \subsetneq G_Z$. Thus, for these graphs, *any* equivariant encoder will produce an "overly self-symmetric" latent embedding. To reconstruct $A$, we must therefore break the latent-space symmetry induced by the equivariant encoder.[8]

In our experiments, we followed the training hyperparameters of Satorras et al. (2021), but trained for fewer epochs (50) using their "`erdosrenyinodes_0.25_none`" dataset and the "`AE`" architecture. Moreover, we follow the setup of Satorras et al. (2021) and break symmetries at the input, but later compare to breaking symmetries of the embedding (post-encoding) as well. Satorras et al. (2021) suggest doing this with random noise as initial node features. However, this breaks *all* permutational symmetry, not just the graph's automorphism group. In contrast, our method is still equivariant to permutations of nodes that are automorphically equivalent. Although it is not feasible to exactly sample from the graph's automorphism group due to computational constraints, one can still hope to sample from a subgroup of $S_n$ containing the automorphism group (i.e., an equivariant variable $Z$ small support).

To apply SymPE, we must decide how to sample $\tilde{g}$. We compute nodewise embeddings using one graph convolutional layer, then let $\tilde{g}$ be a permutation that sorts the scalar embeddings. This method empirically works the best, but we also compare to a heuristic without learned parameters based on Laplacian positional encodings. For this heuristic, we truncate Laplacian positional encodings to the fourth largest singular values, $P \in \mathbb{R}^{n \times 4}$, and then apply a learnable dimensionality reduction $w \in \mathbb{R}^4$ to obtain the vector $y = Pw \in \mathbb{R}^n$. $\tilde{g}$ is obtained by sorting $y$, letting the sorting algorithm break ties, and the symmetry breaking input $\tilde{v} = \tilde{g}v$ is obtained by correspondingly sorting a learned vector $v \in \mathbb{R}^n$. The symmetry breaking input $\tilde{g}v$ is again obtained by sorting $v$, letting the sorting algorithm break ties. Although the Laplacian eigenvectors are only determined up to sign (if the singular values are unique), and have further rotational ambiguity under repeated singular values, sampling from the possible space of such spectral encodings should still constitute an equivariant random variable $Z$.

We also include some additional baselines in Table 5, none of which work as well as our proposed method. In both this table and in the main body, % Error is the same metric as used in Satorras et al. (2021).

---

[7]The possible stabilizers of $\mathcal{Z}$, the latent space of node-wise featurizations, are groups of the form $S_{i_1} \times \ldots S_{i_k}$, where $i_1 + \cdots + i_k = n$.

[8]Of course, one could also use a different latent space, such as a latent space of matrices. However, this may defeat the purpose of learning an expressive, dimensionality-reduced latent space.

Table 5: Full results of cross-entropy loss and reconstruction error

| Method | BCE | % Error | # Param. |
|---|---|---|---|
| Uniform (embedding) | 20.2 | 5.8 | 88,043 |
| No SB (without input encoding) | 19.03 | 6.4 | 88,017 |
| SymPE with Laplacian, only 1 channel | 13.2 | 2.8 | 88,239 |
| Noise (input) | 10.1 | 2.3 | 88,017 |
| No SB (with input encoding for fair comparison) | 10.0 | 3.9 | 101,074 |
| Noise (embedding) | 9.7 | 3.8 | 101,075 |
| SymPE with Laplacian | 7.0 | 1.5 | 88,885 |
| SymPE with Laplacian *and* noise (embedding) | 6.7 | 1.5 | 88,891 |
| Uniform (input) | 5.7 | 1.3 | 88,890 |
| Laplacian (directly input) | 5.6 | 0.010 | 88,885 |
| SymPE with learned features (ours) | **3.7** | **0.078** | 101,750 |

