# OpenReview forum: "Improving Equivariant Networks with Probabilistic Symmetry Breaking"
_ICLR.cc/2025/Conference — ICLR 2025 Poster_

### Official Review · Reviewer_JR9n · 2024-10-21

**Soundness:** 3
**Presentation:** 1
**Contribution:** 3
**Rating:** 6
**Confidence:** 5

**Summary:**

The paper introduces a method for symmetry breaking in equivariant models, particularly focusing on equivariant GNNs. The motivation and relevance of this work are clear. However, while the idea and motivation are compelling, the paper itself is disorganized, unclear, and lacks important details. As it stands, I cannot recommend its acceptance, and significant revisions will be necessary for it to be considered.

**Strengths:**

The paper presents a solid theoretical foundation on the subject and offers a well-motivated experiments section. The proposed method is supported by both theoretical insights and detailed experimental results.

**Weaknesses:**

The presentation of the paper is very poor. From the background through to the experiments, the notations and mathematical explanations are unclear throughout.

**Questions:**

The overall structure of the paper is very poor, and I will highlight the most critical issues in each section.

**Contribution**
In the contribution section, you need to clearly state your achievements. However, you provided an outline of the paper’s structure instead, which is typically referred to as an arrangement.

**Background and main text**
1. Are your distributions discrete or continuous? This distinction needs to be clearly stated, as it drastically changes the mathematics. Since you condition on $X=x$, it suggests discreteness, yet many of your sets appear infinite.
2. What are the input and output random variables, X and Y? What are their input and output spaces? This is very unclear.
3. There are no meaningful references to figures in the paper.
4. The figure legends are unclear.
5. In Equation 2, line 137, Y is a random variable, but your function f maps elements into $Y$. How does that make sense? Also, it’s unclear whether the probability is over X or over epsilon.
6. In Theorem 3.4, equation 3, it’s not clear what the probability is being taken over.
7. In the paragraph beginning at line 235, how do you compute this argmin efficiently over infinite groups or over the set of permutations?
8. In my opinion, Section 5, which extends the discussion to noise, should be moved to the appendix, while expanding your existing work in the main body would be more valuable.
9. Why is Theorem 6.1 necessary?

---

> ### Author Response · Authors · 2024-11-20
> **Response to reviewer JR9n (1/3)**
>
> We thank the reviewer for their time, and for their valuable feedback on our work. We are happy to see that they find the motivation and relevance of the paper clear. We also appreciate that they have found our theoretical results solid and insightful as well as our experiments detailed.
>
> We will respond to each point individually below, and hope to resolve some of the reviewer’s concerns regarding presentation.
>
> To begin, we hope to clarify the reviewer’s questions on probability and notation.
>
> ## Question 1: Are distributions discrete or continuous?
> In our paper, they can be either! For example, $X$ and $Y$ can be token sequences from a finite vocabulary (discrete), or graphs with real-valued node features (continuous). Happily, our theoretical  results really do not depend on which of the two is the case. The appropriate notion of conditional distribution which encapsulates both cases is called [regular conditional probability](https://en.wikipedia.org/wiki/Regular_conditional_probability), and it gives the standard formulas for both conditional probability mass functions and conditional densities (or just [“conditional probability distributions”](https://en.wikipedia.org/wiki/Conditional_probability_distribution)).
>
> ## Question 2: What are the input and output random variables, X and Y? What are their input and output spaces?
>
> As mentioned at the beginning of the preliminaries section (lines 87 and 88), the input and output spaces are simply two sets on which we have probability distributions, and on which the action of $G$ satisfies some basic conditions (see Footnote 1) — so, technically, they should be measurable spaces, which we have added to the paper. In machine learning applications, one generally encodes inputs and outputs as vectors, which would make $\mathcal{X}$ and $\mathcal{Y}$ vector spaces. Again, a realistic example might be the vector space of point clouds in 3D.
>
> ## Question 5: In Equation 2, line 137, Y is a random variable, but your function $f$ maps elements into $\mathcal{Y}$. [...] it’s unclear whether the probability is over $X$ or over $\epsilon$.
>
> As a reminder, Equation 2 was:  $Y =_{a.s.} f(X,\epsilon)$. Here, “a.s.” denotes that the equality holds [almost surely](https://en.wikipedia.org/wiki/Almost_surely), i.e. with probability 1. What does this mean? $X$, $Y$ and $\epsilon$ are all random variables, taking values in the sets $\mathcal{X}$, $\mathcal{Y}$ and $[0,1]$ respectively, as described in the paper. (See the preliminaries paragraph (line 95), where we specified this for $X$ and $Y$.) The fact that $\epsilon$ is a random variable is implied by the fact that we specify its sampling distribution, but we have now stated this fact explicitly to make sure it’s clear. The equation means that, when you sample each of $X$, $Y$, and $\epsilon$, the probability that the random variables satisfy $Y = f(X,\epsilon)$ is equal to 1. One could say the probability is thus “taken over” $X$, $Y$, and $\epsilon$ (but we emphasize that no integration/expectation is being taken, as the term “taken over” might suggest). We have added an explanation of the “a.s.” notation to the preliminaries.
>
> ## Question 6: In Theorem 3.4, equation 3, it’s not clear what the probability is being taken over.
>
> Equation (3) is $Y =_{a.s.} f(X,\tilde{g},\epsilon)$. The explanation is analogous to the one we gave above, except that $\tilde{g}$ is now an additional random variable sampled from the inversion kernel. We emphasize again that “almost surely” means with probability 1, with respect to the joint distribution of all the random variables.
>
> Overall, we have aimed to use notations and concepts that are standard in probabilistic machine learning. These are presented for example in textbooks such as Durrett’s “Probability: Theory and Examples”, and used in prior papers in the area such as [4]. However, we have now additionally made sure to explicitly define the concepts above.

---

> > ### Author Response · Authors · 2024-11-20
> > **Response to Reviewer JR9n (2/3)**
> >
> > We now address the reviewer’s concerns on content.
> >
> > ## Question 3: There are no meaningful references to figures in the paper.
> >
> > We list below the line where every figure is referred to in the text, quoting the relevant line itself for convenience. If there are any specific additions or references the reviewer would find helpful, we are happy to discuss them.
> > * Figure 1 (examples of applications that require symmetry breaking): *Line 048*, “In fact, self-symmetry arises in a variety of applications, often with more complex groups—e.g. non-trivial graph automorphisms, Hamiltonians of physical systems with symmetries, or rotationally symmetric point-clouds (Fig. 1)”
> > * Figure 2 (examples of how a group acts on distributions): *Line 098*, “The action of $G$ on $\mathcal{Y}$ naturally gives rise to an action on distributions, defined by $g \cdot P (Y) ≡ P (gY)$ as shown in Figure 2”
> > * Figure 3 (commutativity diagram indicating how equivariant distributions allow one to break symmetry): *Line 122*, “A key insight is that this does not need to hold for individual samples from $P (Y |X = x)$, as shown in Figure 3.”
> > * Figure 4 (illustration of method): *Line 213*, “Following Theorem 3.4 and as shown in Fig. 4, we propose to represent equivariant conditionals using an equivariant neural network f, and pass in $(x, \tilde{g})$ as input (and $\epsilon$, if so desired).”
> > * Figure 5 (Phase diagrams for the Ising model experiment): *Line 516*, “A more in-depth understanding of the baselines’s shortcomings can be obtained from the predicted phase diagrams (Fig. 5). Spin systems show phase transitions depending on Hamiltonian parameters, similar to molecular systems (see Appendix F)...” [more explanation follows in the next sentences]
> > * Table 1 (graph autoencoding results): *Line 424*, “Breaking symmetries via our method achieves the lowest error (Table 1)”
> > * Table 2 (molecular generation results): *Line 457*, “Results for QM9 are shown in Table 2 and results for MOSES in Table 4 (Appendix G)”
> > In each case, we believe the figure plays an important explanatory role, and supplements the text in which it is mentioned.
> >
> > ## Question 4: The figure legends are unclear.
> >
> > Can the reviewer please clarify which figures are unclear and how they could be improved?
> >
> > ## Question 7: In the paragraph beginning at line 235, how do you compute this argmin efficiently over infinite groups or over the set of permutations?
> >
> > This is a good question! You can find our discussion in the following sentences (lines 241-243). In particular, we reference Appendix C, dedicated to answering this very question. We now summarize the contents of Appendix C. Recall that for small discrete groups (acting on e.g. images and point clouds), one can leverage any existing energy-based canonicalization network, simply by taking the argmin of its output logits (one logit per group element). We have referenced the original papers in which these methods were proposed [5,6]. Of course, this strategy fails for permutation groups. Luckily, for $S_n$ acting on sets, one can cast the energy-based optimization as a *sorting* problem. Here, sampling from the inversion kernel amounts to sampling a permutation that sorts a scalar featurization of the input (again, please see Appendix C for details).
> >
> > We also mention in Appendix C that for continuous groups, one could use Langevin sampling, as well as application-specific heuristics (such as the one we note at the end of the section for point clouds); we do not emphasize this in the main text, as it is not part of our chosen applications.
> >
> > Moreover, Proposition 5.1 states that one need not sample precisely from the inversion kernel (which is provided by the argmin); instead, sampling any equivariant variable $Z$ suffices, although learning will likely be more sample efficient when $Z$ is as close as possible to the proper inversion kernel.

---

> > > ### Author Response · Authors · 2024-11-20
> > > **Response to Reviewer JR9n (3/3)**
> > >
> > > ## Question 8: In my opinion, Section 5, which extends the discussion to noise, should be moved to the appendix, while expanding your existing work in the main body would be more valuable.
> > >
> > > We view Section 5 as a crucial part of our paper.  First, it provides valuable context for our results, by relating them to the existing idea of noise injection. More importantly, we are also able to provide a novel perspective on noise injection, by showing when it can be used to represent any equivariant distribution, which we do not believe was previously known. As noted in lines 382-394, this has useful implications for the expressivity of prior work. Finally, Section 5 provides essential justification for a variety of heuristics one may wish to use, if exactly sampling from the inversion kernel $\tilde{g}$ is inconvenient or computationally expensive, such as the example of graphs in the previous response. There, canonicalizing graphs would solve the graph isomorphism problem, but sampling from a general distribution over permutations that is equivariant with respect to the input is feasible by any variety of methods, such as those we employ in our experiments. Even without the possibility of a computational barrier, Section 5 is also necessary justification for the convenient heuristic for point clouds mentioned in Appendix C, lines 1065-1067. While we expect sampling from the precise inversion kernel to provide greater sample efficiency from a learning perspective, Proposition 5.1 shows that Theorem 3.4 can be relaxed in practice.
> > >
> > > We also note that Section 5 takes up fewer than 30 lines. However, we agree that expanding our discussion of earlier elements of the paper is also valuable. Using extra space from Reviewer 1aaz’s suggestion to move Proposition 4.1 out of the main body, we have added additional clarifications based on the reviewer’s prior feedback as discussed above.
> > >
> > >
> > >
> > > ## Question 9: Why is Theorem 6.1 necessary?
> > >
> > > Theorem 6.1 provides a formal argument that symmetry-breaking should lead to improved generalization. In other words, Theorem 6.1 theoretically justifies the intuition we provided in the introduction, for why one should consider using symmetry-breaking methods (including SymPE) in the first place, as opposed to generic networks. In particular, it shows that in the described setting — invariant $P(X)$ and equivariant $P(Y|X)$ — using our symmetry-breaking model, with an equivariant $f$, *guarantees* better generalization than using an arbitrary $f$ with noise outsourcing to represent the distribution of $Y$. Note that the theoretical result is directly supported by our supplementary experiments on spin models (Appendix F, Table 3).
> > >
> > > ## In the contribution section, you need to clearly state your achievements. However, you provided an outline of the paper’s structure instead, which is typically referred to as an arrangement.
> > >
> > > Indeed, the intention of this paragraph was to simultaneously provide both an outline of the paper and a summary of our contributions. Each sentence states a new contribution, for which we provide the evidence in a section of the paper. Here is what we view as our list of contributions:
> > > * Theorem for representing any equivariant conditional distribution using canonicalization (Section 3)
> > > * Implementation of this theorem via symmetry-breaking positional encodings (Section 4)
> > > * More general theorem for representing any equivariant conditional distribution using equivariant noise injection (Section 5)
> > > * Theorem on the generalization benefits of breaking symmetries (Section 6)
> > > * By application of Section 6, proof of expressivity of prior work (Section 7, which also includes other related work)
> > > * Experimental validation (Section 8)
> > >
> > > We are open to any specific suggestions from the reviewer on this point.
> > >
> > > Please let us know if we can answer any other questions. Note that the changes in the revised manuscript have been highlighted in blue for clarity.
> > >
> > > [1] Petrus Mikkola, Milica Todorovic,  Jari Järvi,  Patrick Rinke, Samuel Kaski. Projective Preferential Bayesian Optimization. ICML 2020.
> > >
> > > [2] Steffen Grünewälder, Guy Lever, Luca Baldassarre, Sam Patterson, Arthur Gretton, Massimilano Pontil. Conditional mean embeddings as regressors. ICML 2012
> > >
> > > [3] Eugenio Clerico, Amitis Shidani, George Deligiannidis, Arnaud Doucet. Chained Generalisation Bounds. COLT 2022.
> > >
> > > [4] Benjamin Bloem-Reddy, Yee Whye Teh. Probabilistic Symmetries and Invariant Neural Networks. JMLR 2020.
> > >
> > > [5] S.-O. Kaba, A. K. Mondal, Y. Zhang, Y. Bengio, and S. Ravanbakhsh. Equivariance with learned canonicalization functions. ICML 2023.
> > >
> > > [6] J. Kim, D. Nguyen, A. Suleymanzade, H. An, and S. Hong. Learning probabilistic symmetrization for architecture agnostic equivariance. NeurIPS 2024.

---

> ### Comment · Area_Chair_buA7 · 2024-11-25
> **The author-reviewer discussion period is ending soon**
>
> Dear reviewer,
>
> Please engage in the discussion as soon as possible. Specifically, please acknowledge that you have thoroughly reviewed the authors' rebuttal and indicate whether your concerns have been adequately addressed. Your input during this critical phase is essential—not only for the authors but also for your fellow reviewers and the Area Chair—to ensure a fair evaluation.
> Best wishes,
> AC

---

### Official Review · Reviewer_XGk3 · 2024-10-31

**Soundness:** 3
**Presentation:** 3
**Contribution:** 3
**Rating:** 8
**Confidence:** 4

**Summary:**

The paper addresses the inability of equivariant neural networks to break symmetries in their outputs, which is a limitation for certain tasks. The authors present a theoretical framework for representing equivariant conditional distributions, allowing symmetry breaking via randomized canonicalization. They propose SymPE (Symmetry-breaking Positional Encodings), a method that expands the representational power of equivariant networks while retaining symmetry inductive bias. Experimental results demonstrate that SymPE significantly improves performance across various applications, including diffusion models for graphs, graph autoencoders, and lattice spin system modeling.

**Strengths:**

1. The paper considers an important but challenging task.
2. The authors provide SymPE, a method motivated by both theoretical and empirical studies.
3. Even though the paper has a theoretical contribution, it is not difficult to read.

**Weaknesses:**

Actually, the paper is solid and interesting. The key limitation admitted by the authors is that the proposed method requires access to a canonicalization method, and more general theoretical results are also desirable.

**Questions:**

I have no questions.

---

> ### Author Response · Authors · 2024-11-20
> **Response to Reviewer XGk3**
>
> We sincerely thank the reviewer for their thoughtful review and appreciation of our work. We are happy that they felt that our paper was easy to read, that it addresses an important and challenging problem, and that we propose a well-motivated method backed by experiments and theory.

---

### Official Review · Reviewer_1aaz · 2024-11-03

**Soundness:** 3
**Presentation:** 3
**Contribution:** 3
**Rating:** 8
**Confidence:** 4

**Summary:**

The paper proposes to sample a group element probabilistically to break the self-symmetry of an input to an equivariant network. This allows more flexible model outputs with lower symmetry, which is crucial for prediction and generation tasks with self-symmetries. The group element is sampled from an inversion kernel, which can be thought of as shifting the uniform distribution on the stabilizer of a data point $x$ by a learned canonicalization element $g = \gamma(x)$. The network then takes the randomly sampled $g$ as input in addition to $x$ to break symmetry.

**Strengths:**

The main idea of this paper is to take a randomly sampled group element as an additional input to the equivariant network to break the self-symmetry of an input. The proposed method is straightforward yet effective, tested on various tasks including graph reconstruction and generation, and the ground-state prediction of Ising models. The authors provide theoretical justifications for the method, explaining the motivation of key design clearly. There are also very helpful discussions on alternative methods to break symmetry, e.g. injecting totally random noise instead of that from a lower entropy.

**Weaknesses:**

* The analog to the positional encoding in Transformers is a bit confusing to me. In the setup of this paper, we want to break the **self-symmetry** of the data $x$, i.e. $gx=x$. However, in terms of sequence modeling, there is no self-symmetry in the data, i.e. if we permute the words in a sentence it would no longer be the same sentence. It is the symmetry of the **model** that we want to break. While you could still argue that your method also breaks the symmetry of the model in the sense that $f(gx) = gf(x)$ is not always true, in essence your method is still equivariant if you consider the joint action on the sampled group element. Also, the word "position" actually has different meaning in these two contexts. The positions in the Transformer PE refer to the indices of words in a sequence, i.e. **in** a single data point. On the other hand, in your method, the positions refer to the group element sampled from the inversion kernel, i.e. the hypothetical position **of** a data point on the group action orbit. I feel this is quite different from its literal meaning in the Transformer PE, and I don't feel this analog provides a good intuition.
* Regarding the encoding of the group element, I wonder if it's better to define things in the language of group representations, since it's already based on the assumption that $G$ acts freely on $v$ by matrix multiplication. Also, I feel the explanation in the text is good enough and Prop 4.1 does not provide any new information.
* Consider changing some notations, e.g. in L275 $e \leftarrow \tilde g v$, $e$ may be confusing because it can also refer to the identity element of the group.
* The experiment section could benefit from more clarifications. See questions.

**Questions:**

* Eq (1): should it be this way or $P(Y | X = g^{-1}x) = P(gY | X = x)$?
* Why are the results in Table 1 different from those in Satorras et al (Figure 5)? From my understanding, the "No SB" row in your table refers to an equivariant GNN without symmetry breaking and should match the EGNN performance reported by Satorras et al. If there are differences in settings that cause the discrepancy, they should be highlighted.
* In Table 2, apart from NLL, other metrics are more similar across different models. How do you interpret this? Also, DiGress + SymPE (nodes only) has better validity vol. stability scores but is not highlighted. Is this intentional or a mistake?
* L500: can you specify the action of $g \in p4m$ on $(J, h, \sigma)$?
* L508: What does it mean by "Hamiltonian parameters are randomly rotated by 90 deg"?
* Does your method have an advantage over G-CNN w/o symmetry breaking in predicting ground-states of Ising models partly because the interaction $J$ and the transverse field $h$ are initialized with self-symmetry? Would the results be different (maybe better for the baseline) if $J$ and $h$ are not constant on the lattice?
* L1259: "symmetry breaking elements are computed through a deterministic canonicalization". If the group elements are deterministic, how do you break symmetry?

---

> ### Author Response · Authors · 2024-11-20
> **Response to Reviewer 1aaz (1/2)**
>
> We sincerely thank the reviewer for their thoughtful and detailed feedback. We are glad that they have found our paper clear, the theoretical results helpful, and the proposed method effective. We respond individually to their points below.
>
> ## Positional encoding in Transformers versus our method
> We agree that there are some subtleties to this suggestive choice of terminology, and thank the reviewer for insightfully highlighting them.
>
> ### Breaking model vs input symmetries
> First, the reviewer is correct that the primary goal of positional encodings (PEs) in transformers is not to break permutation symmetries of the input, but in the model architecture itself. This is part of why we call ours *“symmetry-breaking”* positional encodings, rather than ordinary positional encodings. Nonetheless, in both cases (as the reviewer points out), we are using some kind of positional encoding to turn an equivariant learning method into a less equivariant learning method. However, the analogy runs deeper than this -- we’ll dive into this in the subsequent paragraphs.
>
> ### Relative positional encodings
> Work on relative positional encodings [1] has argued that some degree of inductive bias towards equivariance, such as sequence translations, is desirable. Imagine then that we want our language model to be equivariant to translations (in an idealized case of zero-padded sequences). One could no longer use ordinary positional encodings, which *fully* break permutational symmetry (because they are constant with respect to input permutation, indeed they are independent of the input). Instead, one must use input-dependent positional encodings, which are *equivariant* with respect to input translation. (Then, even if we translate the input sequence, the positional encodings will translate too.) We would therefore apply SymPE  with respect to the **translation group, not the symmetric group**. This would make the full model (SymPE + Transformer) equivariant to only translations, achieving the correct level of equivariance.
>
> ### The two meanings of “position” then coincide
> Seen in this way (above), the two meanings of position as "position of the tokens in the sentence" and "position of the data sample in the orbit" coincide. This is because specifying a single element of the translation group (via sampling from the inversion kernel), specifies the position of all the tokens --- since the relative ordering is preserved by translations. We think that the analogy is therefore still correct in this setting.
>
> However, given the subtleties discussed above, the reviewer makes a fair point that this sequence model example could be more intuitive.
>
> ### Graph positional encodings
> The example of PEs in graphs is more intuitive. In this setting, the main use of PEs is to break symmetries’(automorphisms) and increase model expressivity. Like for sequences, we also have that the group element (permutation) obtained at a sample (graph) level completely specifies the position of individual components (nodes). , Relatedly, the difficulty of designing PEs for graphs has often [2,3] been attributed to the lack of a canonical ordering for this data structure. In SymPE, this problem is tackled directly by learning and sampling a canonical ordering from an inversion kernel. (Or, more realistically, a more general equivariant distribution as in Proposition 5.1.)
>
> Based on the reviewer’s feedback, we have updated our motivation for the term “symmetry-breaking positional encodings”, at the bottom of page 5, in terms of graph positional encodings.
>
> ## Encoding of the group element
> It is true that things could be framed in the language of group representation for this part. However, since most of the results of the paper do not assume that we deal with representations and apply to more general actions, we have preferred to keep the formalism uniform.
>
> The reviewer’s other suggestion to remove Prop 4.1 from the main text is a good idea, and we have implemented it (moving Prop 4.1 to the appendix).
>
> ## Notation for encoding vector
> This is a good suggestion. We have implemented it and changed the notation for the positional encoding from $e$ to $\tilde{v}$.
>
> ## Eq (1): should it be this way or $P(Y|X=g^{−1}x)=P(gY|X=x)$?
> By $P(gY|X=x)$, what we mean is $P(gY=y|X=x)$, i.e. $P(gY=y|X=x)=P(Y=y|X=gx) $ for all $y$ (interpreted as probability density functions).
>
> ## Different results from Satorras et al. 2021
>
> Thanks to the reviewer for pointing this out. Indeed, although we used the codebase of Satorras et al, the experimental setup was slightly different (we use the “erdosrenyinodes_0.25_none” dataset, the “AE” architecture, and train for fewer epochs) -- but the comparisons within our experiments should be fair. We will clarify these distinctions in the revision.

---

> > ### Author Response · Authors · 2024-11-20
> > **Response to Reviewer 1aaz (2/2)**
> >
> > ## Evaluation of molecular generation through NLL and other metrics
> > This is a good comment, and it is related to the complex problem of evaluation of generative models. The NLL (on test samples) evaluates how well the generative model captures the distribution. However, the other metrics evaluate how well the generated samples conform to chemical validity rules. The two are related but do not necessarily coincide, in the same way that LLM perplexity does not necessarily correspond to accurate answers. One hypothesis for why the NLL might be improved significantly more than other metrics is that the value of the other metrics (except for molecular stability, which SymPE improves) is already very high, which we know because they are close to the dataset value. We have added some discussion of this in the main text.
> >
> > ## Action of the group on the spin Hamiltonian
> > The action of the p4m group on the Hamiltonian is given by translating, reflecting or performing 4-fold rotations on the spin grid and then applying the Hamiltonian function. This is similar to the action of this group on an image (grid of pixels). We have added more mathematical details on this in Appendix F. By “randomly rotating the Hamiltonian parameters by 90 degrees”, we meant that with equal probability we either act on the Hamiltonian via $g=90\deg$, or we don’t do anything ($g=e$). This is a way to augment the test set common in evaluating equivariant models. We have also clarified this in Appendix F thanks to the reviewer’s helpful question.
> >
> > ## $J$ and $h$ being initialized with self-symmetry
> > This is a great question. We indeed think that the self-symmetry of $J$ and $h$ are part of the reason behind the improvement of our method over G-CNNs. This is directly related to the symmetry breaking problem: G-CNNs cannot break input symmetries, while our method does it easily. It is therefore expected that the results will be better for the baseline for non-symmetric $J$ and $h$. Note that the setup considered in the paper is not artificial, however: the Ising model and its variants assume symmetric J and h, yet are still non-trivial and of high interest.
> >
> > ## Deterministic canonicalization
> > The use of a deterministic canonicalization (instead of uniform sample from inversion kernel) will actually still lead to symmetry breaking, in the sense that even if $\tilde{g}$ is a deterministic function of $x$, $f(x,\tilde{g})$ can still be less symmetric than $x$ itself. However, the different ways in which the symmetry can be broken will not be sampled uniformly (resulting in a deterministic function of $x$ rather than an equivariant distribution, if $\epsilon$ is omitted). Taking the example from Figure 1 of learning a function that maps benzene to  dichlorobenzene, this would correspond to correctly outputting dichlorobenzene (and breaking the symmetry), but always with the same orientation. For the Ising model experiment, deterministic or stochastic canonicalization does not make any difference in the metrics, since all they measure is if one of the possible ground-states was obtained. However, in light of your question and for consistency with the rest of the results, we have modified our implementation to use a stochastic canonicalization and rerun the experiments. The results are the same, and we have updated the explanation of Appendix F accordingly.
> >
> > We again thank the reviewer for their excellent suggestions, which we think have helped us improve the paper. Note that the changes in the revised manuscript have been highlighted in blue for clarity. Please let us know if we have addressed all your concerns appropriately.
> >
> > [1] P. Shaw, J. Uszkoreit, and A. Vaswani. Self-attention with relative position representations. arXiv preprint arXiv:1803.02155, 2018.
> >
> > [2] V. P. Dwivedi, A. T. Luu, T. Laurent, Y. Bengio, and X. Bresson. Graph neural networks with learnable structural and positional representations. In International Conference on Learning Representations, 2022.
> >
> > [3] L. Ma, C. Lin, D. Lim, A. Romero-Soriano, P. K. Dokania, M. Coates, P. Torr, and S.-N. Lim. Graph inductive biases in transformers without message passing. In International Conference on Machine Learning, pages 23321–23337. PMLR, 2023.

---

> > > ### Comment · Reviewer_1aaz · 2024-11-20
> > > **Response to authors**
> > >
> > > Thank you for the detailed response. I think the updated manuscript clearly explains the intuition of symmetry-breaking positional encoding. My other concerns have also been addressed. Therefore, I will raise my score to 8.

---

### Official Review · Reviewer_oSLY · 2024-11-04

**Soundness:** 3
**Presentation:** 3
**Contribution:** 3
**Rating:** 6
**Confidence:** 3

**Summary:**

The paper introduces a new method for breaking symmetries in equivariant neural networks through the probabilistic symmetries concept and randomized canonicalization. The proposed approach is implemented as a sampling from a canonicalization network, with additional positional encoding (SymPE), along with the equivariant network.

**Strengths:**

The paper shows a theoretical motivation for the proposed method and extends previous works on equivariant distribution to tackle the symmetry breaking case. It also conducts evaluations on three tasks: graph autoencoder (using EGNN), graph generation (on QM9 using DiGress), and ground-states of the Ising model (using G-CNN), with improvement over the equivariant baselines.

**Weaknesses:**

The limitation of the method, as already mentioned, is that it requires sampling from a canonicalization function in addition to using equivariant networks, which might lead to increased complexity and additional computations.

**Questions:**

There are some works on positional encodings for molecules beyond Laplacian, for example (1). Can you comment on that and how it might perform compared to SymPE?

1. Dwivedi, Vijay Prakash, Luu, Anh Tuan, Laurent, Thomas, Bengio, Yoshua, and Bresson, Xavier. "Graph Neural Networks with Learnable Structural and Positional Representations." 2022

---

> ### Author Response · Authors · 2024-11-20
> **Response to Reviewer oSLY**
>
> We thank the reviewer for their positive feedback and valuable suggestions. We address specific questions below.
>
> ## Increased computational cost due to canonicalization
> This is a valid concern. However, to summarize, we have found that this is not an issue in practice.
>
> First, we reiterate why the reviewer’s point is valid. Depending on the group and data type, properly canonicalizing may not even be solvable in polynomial time -- for example, in the case of $S_n$ acting on graphs, it would solve graph isomorphism (since, to determine if two graphs are isomorphic, one need merely canonicalize them both). However, Proposition 5.1 is our saving grace. It states that actually, we do not need a strict canonicalization function to represent any equivariant distribution; instead, any equivariant variable $Z$ will suffice. (The point of using a canonicalization, as discussed in Section 5 and Appendix E, is for learning efficiency -- we wish to inject as little noise as possible into the network.) Therefore, even if we are not exactly canonicalizing, such as using an invariant graph network to output permutations in the graph diffusion experiment (Section 8), we can still break symmetries perfectly.
>
> Moreover, outside of cases like the aforementioned graphs, canonicalization via fixed analytical procedures is a widespread preprocessing step in many fields of science (as just one example, it is standard practice to canonicalize small molecules via SMILES strings using computational chemistry software like rdkit). Therefore, there are often heuristics readily available for canonicalization, or at least for outputting an equivariant $Z$ as described above. For example, at the end of Appendix C, we suggest a simple and efficient heuristic for point clouds based on randomly choosing two points of maximal radius. Our work allows one to harness such procedures for more general (learnable) symmetry-breaking.
>
> Alternatively, if one uses an equivariant network to learn a canonicalization, as introduced by [3], the canonicalization function can be made much smaller than the neural network used for the prediction, resulting in a small additional overhead. Evidence for this is provided in the main text for graph autoencoder experiment and in the appendix for the other experiments: in the case of a learned canonicalization, using SymPE results in a negligible increase in the number of parameters compared to using only an equivariant network. The canonicalization networks are themselves cheap. We have provided explicit evidence for this in Tables 3 and 4 of the revised manuscript by including inference time for the spin experiment, and training time for the graph diffusion model. As shown, the additional computational overhead is relatively low (28h vs 34h for training the graph diffusion models, and 0.49s vs 0.52s for the forward time on the spin experiment).
>
> ## Comparison with other positional encodings
> That is a great comment. We have added some discussion of this work in the revised manuscript, along with other graph positional encodings such as RFP (Random Feature Propagation) [1] and RRWP (Relative Random Walk Position) [2], since our results have important implications in this regard. The core idea is that the learned positional encodings in Dwivedi et al. are initialized from Random Walk Positional Encodings. Although they can improve the expressivity of a GNN, they are still strictly equivariant. Therefore, as highlighted by our theoretical results, they also cannot break symmetries (i.e. nodes in the graph that are equivalent under automorphism will have the same embedding) and do not allow one to represent arbitrary equivariant *distributions*. An important contribution of our work is the proof that for a positional encoding to break graph symmetries, it must be sampled from a distribution that satisfies the conditions of Proposition 5.1.
>
> Please let us know if we have satisfyingly addressed your concerns, and if you have any other questions. Note that the changes in the revised manuscript have been highlighted in blue for clarity.
>
> [1] M. Eliasof, F. Frasca, B. Bevilacqua, E. Treister, G. Chechik, and H. Maron. Graph positional encoding via random feature propagation. In International Conference on Machine Learning, pages 9202–9223. PMLR, 2023.
> [2] L. Ma, C. Lin, D. Lim, A. Romero-Soriano, P. K. Dokania, M. Coates, P. Torr, and S.-N. Lim. Graph inductive biases in transformers without message passing. In International Conference on Machine Learning, pages 23321–23337. PMLR, 2023.
> [3] S.-O. Kaba, A. K. Mondal, Y. Zhang, Y. Bengio, and S. Ravanbakhsh. Equivariance with learned canonicalization functions. ICML 2023.

---

> > ### Author Response · Authors · 2024-11-25
> > **Response follow-up**
> >
> > Dear reviewer,
> >
> > We have tried our best to address your questions (see our rebuttal above), and revised our paper following suggestions from all reviewers.
> >
> > Please let us know if you have any follow-up questions or if you need further clarifications. Your insights are valuable to us, and we stand ready to provide any additional information that could be helpful.

---

> > > ### Comment · Reviewer_oSLY · 2024-11-26
> > >
> > > I thank the authors for the detailed response and their follow-up. I don't have additional questions, and I think my score reflects the paper.

---

### Meta-Review · Area_Chair_buA7 · 2024-12-19

**Metareview:**

This paper presents an intriguing, effective, and theoretically sound method for symmetry breaking in equivariant neural networks; the implementation is quite straightforward—namely, adding a symmetry-breaking positional encoding to a canonicalization sampling procedure. Yet, the resulting method can be shown theoretically to expand the expressiveness of the equivariant neural networks. All the reviewers agree that the proposed approach is both novel and significant. While there were a few minor concerns (unrelated to the core significance of the method or the experimental results), most of them were successfully resolved during the rebuttal. I concur with the reviewers’ assessment and therefore recommend acceptance.

**Additional Comments On Reviewer Discussion:**

None of the reviewers were against acceptance after the author-reviewer discussion period, and none of them changed their minds during the final reviewer discussion phase.

---

### Decision · Program_Chairs · 2025-01-22

Accept (Poster)